# Optimization the stochastic optimal reactive power dispatch with renewable energy resources using a modified dandelion algorithm

Naima Agouzoul[1]*, Aziz Oukennou[2], Faissal Elmariami[1], Mohamed Ebeed[3,4]*, Jamal Boukherouaa[1], Rabiaa Gadal[1], Mokhtar Aly[5], Emad A. Mohamed[6]

**1** National Superior School of Electricity and Mechanics (ENSEM), Hassan II University of Casablanca, Oasis, Casablanca, Morocco, **2** Engineering and Applied Physics Team, Higher School of Technology, Sultan Moulay Slimane University, Beni Mellal, Morocco, **3** Electrical Engineering Department, Faculty of Engineering, Sohag University, Sohag, Egypt, **4** Engineering, University of Jaén, EPS Linares, Jaén, Spain, **5** Facultad de Ingeniería, Arquitectura y Diseño Universidad San Sebastián, Santiago, Chile, **6** Department of Electrical Engineering, College of Engineering, Prince Sattam Bin Abdulaziz University, Al Kharj, Saudi Arabia

\* mebeed@eng.sohag.edu.eg (ME), naima.agouzoul@ensem.ac.ma (NA)

## Abstract

Improvement performance of transmission systems is crucial task that can be boosted via optimal reactive power dispatch (ORPD). However, the continuous variations of load demand and the power produced by the renewable energy sources (RERs) increases the complicities of solving the stochastic optimal reactive power dispatch (SORPD) solution. In this regard, a modified Dandelion Optimizer (MDO) algorithm is introduced to optimize the SORPD solution with taking into consideration the stochastic fluctuations or the random variations of the load demand and the power produced by RERs. The suggested MDO depends upon developing the searching exploration and exploitation abilities by integration of three methodologies involving the Quasi-oppositional-based-learning (QOBL), the Weibull flight motion strategy (WFM) and the fitness distance balance (FDB). The SORPD is solved for IEEE 30-bus system to reduce summation of expected power losses (SEPL) and enhance the summation of expected voltage stability (SEVS) with and without integration RERs. The uncertainties of the load demand and the power produced by the RERs are represented using Monte Carlo simulations and scenario reduction approach in which 15 scenarios are generated to model the stochastic nature of the load demand and the power produced by RERs. The simulation results reveal to that application the proposed algorithm for SORPD can reduce the SEPL and improve SEVS considerably, especially with integration of the RERs. The Comparative results demonstrate that the MDO algorithm is the best for solution the SORPD against sand cat swarm optimization (SCSO), gorilla troop optimizer (GTO), harmony search (HS), and Beluga whale optimization (BWO).

**Data availability statement:** All relevant data are within the manuscript.

**Funding:** The author(s) received no specific funding for this work.

**Competing interests:** No competing interests exist.

## 1. Introduction

Incorporating Renewable Energy Resources (RERs) into the electrical grids is no longer optional, but crucial for fostering a sustainable, resilient, and efficient energy system. The integration of RERs plays a key role in minimizing greenhouse gas emissions, utilizing local resources, and enhancing energy security. However, the inherent variability of RERs, including wind and solar, increases the uncertainties in energy generation, which pose significant challenges to grid planning and management. These fluctuations, driven by environmental conditions like wind speed, solar irradiation, and weather patterns which lead to mismatch or deficit between generated powers and demand.

Rising electricity demand simultaneously compounds these challenges, often leading to power losses and voltage instability which are caused by inaccurate forecasts. Addressing these issues necessitates advanced management strategies to ensure reliable grid operation. This paper focuses on solving the SORPD problem with considering the uncertainties of the RERs and the load demand. The main goal is to diminish power losses, reduce voltage deviations, and enhance stability of the voltage by adjusting generator voltages, transformer tap ratios, and reactive powers of the shunt capacitors.

The electrical system is inherently subject to various uncertainties, particularly in load demand and the RERs. The activities of customers during the day ahead lead to variations in demand. At the same time, the output of photovoltaic (PV) units is influenced by variations of the irradiance and the temperature while the output power of the wind turbine (WT) is affected by wind speed variations. Effectively managing these uncertainties is essential for maintaining optimal system operation.

To assess system performance under uncertainties, researchers utilize the Weibull probability distribution for wind velocity, the lognormal probability distribution for solar irradiance, and the normal probability distribution for load demand, effectively modeling the uncertainties in RERs and load fluctuations. Addressing the uncertainties in SORPD problem solution requires advanced probabilistic and numerical methods. As shown in Fig 1, three widely used approaches to manage these uncertainties including the Monte Carlo simulation, point estimation method, and scenario-based method.

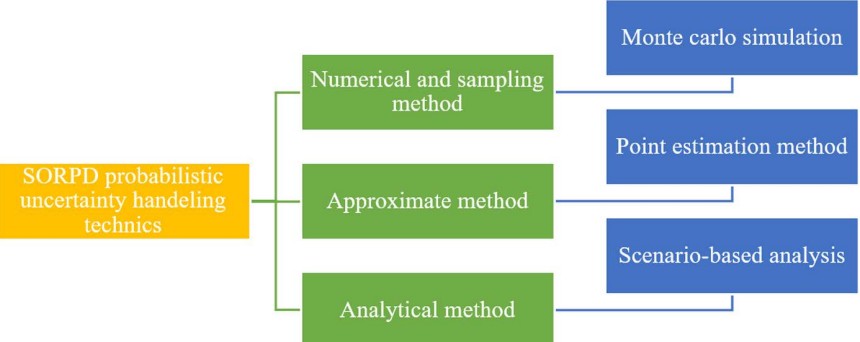

**Fig 1. SORPD probabilistic uncertainty handling techniques.**

Monte Carlo simulation is an efficient method due to its flexibility and precision, as it generates many random samples based on the probability distributions of input variables [1]. This method provides reliable estimates and explores various scenarios even in the absence of historical data, while quantifying variance with high accuracy [2,3]. Despite its high computational cost, it remains an effective and robust probabilistic approach. [4–6]. Point estimation methods, such as the two-point estimation method (TPEM), can quantify uncertainty with fewer samples than Monte Carlo simulation [7]. TPEM approximates the probability distribution using only two deterministic points, reducing the computational burden [8,9]. Although point estimation methods are simple and computationally efficient, they do not capture the full range of uncertainties within the system, which may lead to either overly optimistic or pessimistic decisions. Additionally, this method is sensitive to assumptions and input parameters and may be less accurate than Monte Carlo simulation, particularly for highly nonlinear systems [10]. The scenario-based analysis is a more sophisticated and less computationally intensive approach [11] but has inherent limitations. It relies on a limited number of predefined scenarios, which can lead to less precise representations of uncertainty [9]. The selection process for these scenarios is often subjective, potentially introducing bias and neglecting significant but less obvious cases.

In SORPD, the MCS method is often preferred due to its ability to model and handle complex systems and provide comprehensive insights into uncertainty. Various studies highlight the significance of the MCS in addressing uncertainties associated with load variations and renewable energy integration [12]. This approach enables a comprehensive evaluation of scenarios, supporting the advancement of efficient and sustainable power systems [13].

Recent literature has extensively explored the probabilistic methods and metaheuristic algorithms to solve the SORPD with considering uncertainties of systems including the load demand and the RERs. The MCS method has been applied with different algorithms, such as IMPA [14], NSGA-II [15], Rao-3 [16], and SHADE [17]. The Lognormal, the Weibull, and the normal distributions were used to model uncertainties of the solar irradiance, wind speed, and load demand. Both power loss (PL) and voltage deviation (VD) were optimized, while the Voltage Stability Index (VSI) objective was overlooked. Similarly, in [13] and [18], MCS was used alongside the ToP and NSGA-II, respectively, with the same distributions applied to model uncertainties on the IEEE 30-bus system. However, critical objectives such as PL and VD, essential for the reliability and stability of the distribution network, were notably overlooked in their analyses.

In [19], the SORPD was solved using PSO-CGSA algorithm while MCS is utilized for handling the uncertinity on the IEEE 13-bus system, using beta and normal distributions for solar energy and load uncertainties, respectively. In [20], The EGTO along with MCS were applied to solve the SORPD for the IEEE 30-bus system with considering the load and the PV generation while the generated power by wind energy weren't taken into consideration.

In [21], the MCS method was used along with the NSGAII algorithm for SORPD solution of the IEEE 30-bus system, integrating lognormal and Weibull distributions. Nevertheless, this approach demonstrates inadequate consideration of the complexities inherent in real-world systems, such as load uncertainty, and fails to optimize the VSI. In [22], the QPSODM algorithm was applied along with the MCS method for solving the SORPD of the 14-bus IEEE system and the isolated power network of Adrar. Nevertheless, this method overlooked VSI. Similarly, in [23], the modified jellyfish optimizer (MJSO) was implemented with the MCS method for SORPD of the IEEE 30-bus system. Finally, in [24], the MCS was applied with the NSGA-II algorithm for SORPD solution utilizing the lognormal, the Weibull, and the normal distributions on the same system; however, the power losses (PL) were not addressed.

The PEM has been widely used in various studies focusing on SORPD. In [25], the lexicographic optimization approach, combined with the enhanced weighted ε-constraint hybrid approach, was employed to model wind uncertainty using the Weibull distribution within the IEEE 30-bus system. However, the related uncertainties of solar energy and load, as well as critical objectives like power losses (PL) and the VSI, were inadequately addressed in this study. Similarly, in [26], the interior-point method was implemented on both the IEEE 9 and 30-bus systems, with the Weibull and the normal distributions employed to represent wind and load demand uncertainties, respectively. The PL objective and solar energy

uncertainty were also not considered in this approach. Lastly, in [27], The PCA-RCGA approach was applied for SORPD solution of the IEEE 30-bus system using the Weibull and the normal distributions. However, solar energy uncertainty was inadequately addressed, and the VSI was omitted.

The SBA were investigated in several studies for its application in SORPD. In [28], the FPSOGSA algorithm was used to incorporate wind and load demand uncertainties in the IEEE 30- and 57-bus systems, utilizing Weibull and normal distributions. However, the associated uncertainty with solar energy was inadequately addressed, and the VSI objective was overlooked. In [29], the SBA method was applied using GAMS with the SNOPT and SBB solvers on the IEEE 30- and 118-bus systems. However, solar irradiance uncertainties were not adequately considered. Similarly, in [30], GAMS was used with the SNOPT and SBB solvers for the IEEE 57-bus system. Nevertheless, wind energy uncertainties were not sufficiently addressed. In [31], The Gravitational Search Algorithm and Particle Swarm Optimization (GSA-PSO) were applied to solve the SORPD on the IEEE 30-bus system, utilizing the lognormal, the Weibull, and normal distributions for modeling system's uncertainties. It was found that SBA is less effective than MCS due to limited scenario coverage, leading to potentially suboptimal outcomes. Similarly, in [32], the Artificial Hummingbird Algorithm (AHA) was applied to solve the SORPD of the IEEE 14- and 39-bus systems, using the same distributions while the VSI were neglected which indicates the limitations of SBA in comparison to MCS. Additionally, the Marine Predators Algorithm (MPA) was applied to the IEEE 30-bus system, utilizing beta, Weibull, and normal distributions to model uncertainties. The VSI objective was neglected in this approach [33].

The study of existing work has clearly highlighted the advantages and disadvantages of each probabilistic method. Finally, it appears that the SBA and MCS methods are more widely used compared to the PEM. Since our approach is based on the use of scenarios, a specific comparison between MCS and SBA is presented in Table 1.

In [50], a comparative study between DO, PSO, GWO and other algorithms was carried out on in multidimensional experiments. In this analysis, the DO algorithm was compared with PSO [51], which, although showing a good exploitation capacity [52–54] sometimes suffers from limited exploration and unsatisfactory overall performance. DO was also compared with the GWO algorithm. Although GWO showed better performance in exploration, in particular by avoiding premature convergence and maintaining a diversity of solutions in the search space, it nevertheless has limitations in local exploitation, not always allowing precise refinement of solutions [55–57].

Despite all the limitations identified for PSO and GWO compared with DO, both algorithms outperformed the latter in terms of exploitation and exploration. The study in [50] showed that DO suffer from limited exploration and exploitation capability. To overcome DO's structural limitations, techniques such as Fitness-Distance Balance (FDB) [58], Quasi-Oppositional Based Learning (QOBL) [59], and Weibull Flight Motion (WFM) [60], were integrated to improve its robustness and efficiency. The WFM is used to boost the exploration of the applied optimizer, and its effectiveness has been demonstrated in [60–62]. Similarly, the FDB improves the global search capability, with successful applications reported [63–69].Furthermore, the QOBL can enhance the exploitation process, as depicted on [38,70–75].

The key contributions are summarized as follows:

- A developed Modified Dandelion Algorithm (MDA) is proposed based on the QOBL and the FDB and the WFM strategies.

- The proposed optimizer is applied for solving the stochastic optimal reactive power dispatch of the IEEE 30-bus system with the integration of renewable energy resources.

- The stochastic optimal reactive power dispatch is solved for two objective functions including total expect ed power losses and the total expected voltage stability.

- The stochastic optimal reactive power dispatch is solved by considering the uncertainties of the demand variations and the fluctuations of the generated powers for the PV and wind turbine generation systems.

- The system's performance is analyzed and assessed with and without incorporating renewable energy resources under a set of uncertain scenarios.

- Verifying effectiveness and validity is verified compared to other techniques like Sand Cat swarm optimization (SCSO), Gorilla troops optimizer (GTO), harmony search (HS), and Beluga whale optimization (BWO).

The arrangement of the paper is organized as follows: Problem formulation describes the SORPD problem and its associated objective functions. Uncertainties modeling focuses on modeling uncertainties in solar irradiance, wind velocity, and electricity demand. DO and MDO modeling offers a detailed overview of the Dandelion Optimizer (DO) algorithm, followed by an overview of the proposed Modified Dandelion Optimizer (MDO) approach. Analysis and discussion of simulation

**Table 1. Comparison of the proposed method with existing hybrid optimization approaches based on MCS and SBA methods for solving the SORPD problem.**

| Ref | Optimiza-tion method | Year | The objective functions | | The System | Uncertainties | | | Number of scenarios | Sensitivity Analysis of Scenario Number |
|---|---|---|---|---|---|---|---|---|---|---|
| | | | power loss minimization | voltage stability enhancement | | Solar irradiance | Wind speed | Load demand | | |
| **Monte Carlo Simulation (MCS)** | | | | | | | | | | |
| [34] | MJAYA | 2021 | ✓ | ✗ | 30-bus,57-bus | ✗ | ✗ | ✓ | 5 | ✗ |
| [35] | EGWO | 2019 | ✓ | ✗ | 30-bus | ✗ | ✗ | ✓ | 10 | ✗ |
| [36] | AMRFO | 2023 | ✓ | ✗ | 30-bus, 118-bus | ✗ | ✓ | ✓ | 10 | ✗ |
| [37] | SCENRED2 | 2025 | ✗ | ✗ | Sample microgrid | ✓ | ✓ | ✓ | 10 | ✗ |
| [19] | PSOS-CGSA | 2023 | ✓ | ✗ | 13-bus | ✗ | ✓ | ✓ | 10 | ✗ |
| [20] | EGTO | 2023 | ✓ | ✓ | 30-bus | ✗ | ✓ | ✓ | 10 | ✗ |
| [38] | MRUN | 2023 | ✓ | ✓ | 57-bus | ✓ | ✓ | ✓ | 10 | ✗ |
| [39] | ABWO | 2024 | ✓ | ✓ | 30-bus | ✓ | ✓ | ✓ | 12 | ✗ |
| [40] | SNOPT-SBB | 2017 | ✓ | ✓ | 30-bus | ✗ | ✓ | ✓ | 15 | ✗ |
| [15] | NSGA-II | 2020 | ✓ | ✗ | 30-bus | ✓ | ✓ | ✓ | 15 | ✗ |
| [41] | IGWO | 2024 | ✓ | ✗ | 30-bus | ✗ | ✗ | ✓ | 15 | ✗ |
| [13] | ToP-MOEA | 2023 | ✓ | ✗ | 30-bus | ✓ | ✓ | ✓ | 20 | ✗ |
| [42] | SBB | 2015 | ✓ | ✗ | 30-bus, 118-bus | ✗ | ✗ | ✓ | 21 | ✗ |
| [21] | BiCo | 2023 | ✓ | ✗ | 30-bus | ✓ | ✓ | ✗ | 24 | ✗ |
| [43] | LAPO | 2019 | ✓ | ✗ | 30-bus | ✓ | ✓ | ✓ | 25 | ✗ |
| [16] | Rao-3 | 2021 | ✓ | ✗ | 30-bus, 57-bus, 118-bus | ✓ | ✓ | ✓ | 25 | ✗ |
| [14] | IMPA | 2022 | ✓ | ✗ | 30-bus | ✗ | ✓ | ✓ | 25 | ✗ |
| [44] | MAHA | 2023 | ✓ | ✗ | 30-bus | ✓ | ✓ | ✓ | 25 | ✗ |
| [45] | SHADE-EC | 2019 | ✓ | ✗ | 30-bus, 57-bus | ✓ | ✓ | ✓ | 25 | ✗ |
| **Scenario-Based Approach (SBA)** | | | | | | | | | | |
| [46] | SBB-SNOPT | 2016 | ✓ | ✗ | 30-bus, 118-bus | ✗ | ✓ | ✓ | 15 | ✗ |
| [27] | PCA-RCGA | 2020 | ✓ | ✗ | 30-bus | ✗ | ✓ | ✓ | 15 | ✗ |
| [31] | FPSOGSA | 2020 | ✓ | ✓ | 30-bus | ✗ | ✓ | ✓ | 15 | ✗ |
| [47] | MSA | 2019 | ✓ | ✗ | 30-bus | ✓ | ✓ | ✓ | 25 | ✗ |
| [48] | IMPA | 2022 | ✓ | ✗ | 30-bus | ✓ | ✓ | ✓ | 25 | ✗ |
| [24] | ILAPO | 2020 | ✓ | ✗ | 30-bus | ✓ | ✓ | ✓ | 27 | ✗ |
| [33] | MPA | 2020 | ✓ | ✗ | 30-bus | ✓ | ✓ | ✓ | 27 | ✗ |
| [49] | HFEA | 2020 | ✓ | ✗ | 14-bus | | ✓ | ✓ | 75 | ✗ |
| – | Proposed algorithm | – | ✓ | ✓ | 30-bus | ✓ | ✓ | ✓ | 15 | ✓ |

results presents and analyzes the results of applying the proposed methods to solve the SORPD problem. Finally, Conclusion lists the key findings and conclusions of the paper.

## 2. Problem formulation

The aim of Stochastic Optimal Reactive Power Dispatch (SORPD) is to reduce active power losses while increasing voltage stability in the electrical network. To achieve these objectives, it is necessary to regulate the generator voltages, adjusting the transformer taps to keep voltage levels under control in different areas of the network, and fine-tune reactive power management.

All these adjustments must strictly comply with a set of system-specific rules, including equality and inequality constraints including the power flows, the voltage limits, the reactive generation, the transformer tap positions, and the reactive compensation.

In addition, it is imperative to take into account the uncertainties associated with variations in renewable energy sources and changes in load demand.

As a result, SORPD is considered a complex, non-linear optimization problem.

$$Min \; OF_{un}(s, c) \tag{1}$$

$OF_{un}$ refers to the optimization function, where OFun represents either the expected power losses or the expected voltage deviations and is subject to system equality and inequality constraints, which can be formulated as follows:

$$E_m(s, c) = 0 \; m = 1, 2, \ldots, h \tag{2}$$

$$I_n(s, c) \leq 0 \; n = 1, 2, \ldots, h \tag{3}$$

Where $E_m$ and $I_n$ are the system's equality and inequality constraints.

The state variables ($s$) represent the quantities that characterize the system's operating conditions, which are determined by the values assigned to the control variables. In contrast, the control variables ($c$) refer to the system parameters that can be adjusted to optimize its performance. The main objective of SORPD is to determine the optimal values of the control variables to minimize power losses and improve voltage stability in the presence of system uncertainties while simultaneously evaluating the system's overall performance and ensuring that constraints remains within the acceptable limits. The state and control variable parameter vectors can be defined as follows:

$$s = [P_s, V_L, Q_G, S_T] \tag{4}$$

$$c = [V_G, Q_C, T_p] \tag{5}$$

The slack power bus is denoted as $P_s$, the generation and load buses' voltage magnitude are represented by $V_G$ and $V_L$, respectively. $Q_C$, $T_p$, $S_T$, and $Q_G$ stand for reactive powers of capacitors, taps of transformer, power flow in system, and the injected reactive powers of the generators, respectively.

### 2.1. The objective functions

**2.1.1. Summation of expected power losses (*SEPL*).** The *SEPL* represents the total expected power loss of the power system. by aggregating the power losses across all scenarios, where each Expected Power Loss ($EPL_{scig}$)

corresponding to scenario *scig*. The total number of generated scenarios is denoted by $N_{scig}$.in this paper 13 scenarios have been generated using the MSC methods and scenario based reduction in which the expected power losses are calculated for all scenarios and subsequently summed according (6).

$$SEPL = \sum_{scig=1}^{N_{scig}} EPL_{scig} = \sum_{scig=1}^{N_{scig}} \tau_{S,scig} \times P_{Lss,scig} \tag{6}$$

These transmission line power losses are influenced by various system parameters, including the line conductance $g_{ij}$, the voltage magnitudes $Vi$ and $Vj$ at buses $i$ and $j$, respectively, and the voltage angle difference $\delta_{ij}$. The power losses for each scenario are determined using the following equation:

$$P_{Loss} = \sum_{i=1}^{N_L} g_{ij} \left( V_i^2 + V_j^2 - 2V_iV_j\cos\delta_{ij} \right) \tag{7}$$

where $N_L$ represent the number of transmission lines.

### 2.1.2. Summation of expected voltage stability (*SEVS*) enhancement.

The second objective function aims to enhance the voltage stability of the power network. This is accomplished by decreasing the maximum L-index value at a particular bus within the power system. The *L*-index of a bus quantifies the proximity of that bus to a voltage collapse condition based on the following equation:

$$VS = min\left(L_{max}\right) = min\left(L_{max}\left(L_i\right)\right), i = 1, 2 \ldots, NB \tag{8}$$

where $NB$ is number of buses. The *L*-index $L_i$ for the $i^{th}$ bus is defined as follows:

$$L_i = \left| 1 - \sum_{j=1}^{N_{PV}} Y_{ij} \frac{V_j}{V_i} \right|, \forall i = 1, 2, \ldots, N_{PQ} \tag{9}$$

$N_{PV}$ refers to the number of PV buses, $N_{PQ}$ represents the number of load buses, while $Y_{ij}$ denotes the mutual admittance between bus i and j, which is expressed as follows:

$$Y_{ij} = -[Y_1]^{-1}[Y_2] \tag{10}$$

Here, $Y_1$ and $Y_2$ are submatrices of the system's YBUS matrix, which is derived by isolating the parameters associated with PQ and PV buses. The summation of expected voltage stability (*SEVS*) is calculated according to the following equation:

$$SEVS = \sum_{scig=1}^{N_{scig}} EVS_{scig} = \sum_{scig=1}^{N_{scig}} \tau_{S,scig} \times VS_{scig} \tag{11}$$

where $EVS_{scig}$ presents the expected voltage stability, a term used to express the evaluation of the ability of power systems to maintain stable voltage levels under power system uncertainties. For a given scenario $scig^{th}$, the $EVS_{scig}$ is equal to the voltage stability index weighted by the probability of occurrence of the corresponding scenario.

## 2.2. Constraints

To ensure the integrity of the system's operational parameters and limit the search space, the optimization program incorporates a set of constraints, defined as follows:

**2.2.1. Equality constraints.** The optimal operation of the electrical system depends on compliance with the equations for active and reactive power balance. These equations are formulated as follows:

$$P_{Gi} - P_{Li} = |V_i| \sum_{j=1}^{NB} |V_j| \left( G_{ij}cos\delta_{ij} + B_{ij}sin\delta_{ij} \right) \tag{12}$$

$$Q_{Gi} - Q_{Li} = |V_i| \sum_{j=1}^{NB} |V_j| \left( G_{ij}sin\delta_{ij} - B_{ij}cos\delta_{ij} \right) \tag{13}$$

where, $NB$ refers to the total number of buses in the power network. $B_{ij}$ denotes the susceptance between bus $i$ and bus $j$. $P_{Gi}$ and $Q_{Gi}$ represent the active and reactive power generation, while $P_{Li}$ and $Q_{Li}$ refer to the active and reactive load demand.

**2.2.2. The inequality constraints.** The inequality constraints of the system components are represented by the following inequality constraints:

The generator must remain within the upper and lower limits of active power, reactive power, and bus voltages, which are defined as follows:

$$P_{Gi}^{min} \leq P_{Gi} \leq P_{Gi}^{max}, \forall\, i \in N_G \tag{14}$$

$$Q_{Gi}^{min} \leq Q_{Gi} \leq Q_{Gi}^{max} \tag{15}$$

$$V_{Gi}^{min} \leq V_{Gi} \leq V_{Gi}^{max} \tag{16}$$

The transformer's maximum and minimum tap limits are shown as follows:

$$T_k^{min} \leq T_k \leq T_k^{max}, \forall\, k \in N_T \tag{17}$$

The upper and lower limits of the VAR compensation capacity of shunt capacitor banks are shown below:

$$Q_C^{min} \leq Q_{Cj} \leq Q_C^{max}, \forall\, j \in N_c \tag{18}$$

The apparent power flow on each transmission line must be kept within permissible limits, and the load bus voltage amplitude must be regulated within a tolerable range, as shown below:

$$|S_{Ln}| \leq S_L^{min}, \forall\, n \in N_L \tag{19}$$

The load bus voltage must be kept within an allowable range as follows:

$$V_{Lj}^{min} \leq V_{Lj} \leq V_{Lj}^{max}, \forall\, i \in N_{PQ} \tag{20}$$

 

The number of generator units, transformers, capacitor banks, transmission lines, and the load buses are represented by $N_G$, $N_T$, $N_c$, $N_L$, and $N_{pQ}$, respectively. The active power, reactive power, and voltage at generation units are denoted by $P_G$, $Q_G$, and $V_G$, respectively. Similarly, the capacitor bank's reactive power, transformer tap setting, the apparent power on the transmission line, and load bus voltage are represented by $Q_c$, $T_k$, $S_L$, and $V_{Lj}$, respectively.

Eq. (21) has been developed to efficiently handle these constraints and eliminate all disproportioned solutions based on the weight sum approach:

$$OF_{un} = OF_{uni} + w_{t1} \left( P_{Gi} - P_{Gi}^{lim} \right)^2 + w_{t2} \sum_{i=1}^{N_G} \left( Q_{Gi} - Q_{Gi}^{lim} \right)^2$$
$$+ w_{t3} \sum_{i=1}^{N_Q} \left( V_{Li} - V_{Li}^{lim} \right)^2 + w_{t4} \sum_{i=1}^{N_L} \left( S_{Li} - S_{Li}^{lim} \right)^2 \tag{21}$$

where $OF_{un}$ refer to the objectives function (SEPL, SEVS). $P_{G1}$ is the power generated on the slack bus, $Q_{Gi}$ represents the reactive power output of generating units, and $S_{Li}$ represents the apparent power transmitted through the transmission line. And Lim represents lower and upper limits. variable limits. Whereas $w_1$, $w_2$, $w_3$, and $w_3$ are the penalty weighting factors. These factors have values of 100, 100, 1000, and 100, respectively based on the paper [48].

## 3. Uncertainties modeling

The SORPD is solved with considering uncertainties of solar radiation, wind speed, and loading. These parameters are represented through three probability density functions (PDF) including the Lognormal, the Weibull, and the normal PDFs as follows:

### 3.1. Solar irradiance uncertainty modeling

The uncertainty of solar irradiance is represented using the Lognormal probability density function $f_{sr}(G_{sr})$ which can be represented as follows [76]:

$$f_{sr}(G_{sr}) = \frac{1}{G_{sr} r \sigma_{sr} \sqrt{2\pi}} exp \left[ -\frac{(ln(G_{sr}) - \mu_{sr})^2}{2\sigma_{sr}^2} \right] \quad G_s > 0 \tag{22}$$

where $\mu_{sr}$ is the mean value, and $\sigma_{sr}$ is the standard deviation of the irradiance. The generated power of the PV system can be determined according to the following equation [77]:

$$P_{PV}(G_{sr}) = \begin{cases} P_{Sr} \left( \frac{G_{sr}^2}{G_{std} \times X_c} \right) \text{ for } 0 < G_{sr} \leq X_c \\ [5pt] P_{Sr} \left( \frac{G_{sr}}{G_{std}} \right) \text{ for } G_{sr} \geq X_c \end{cases} \tag{23}$$

where $P_{sr}$ and $P_{PV}$ are the PV system's rate and output power respectively. $G_{sr}$ and $G_{std}$ are the current and the standard solar irradiances, respectively.

### 3.2. Wind speed uncertainty modeling

Variation of wind speed is represented through the Weibull probability density function. $f_{v_{wd}}(v_{wd})$ which can be expressed as follows [13]:

$$f_{v_{wd}}(v_{wd}) = \left( \frac{\beta_{wd}}{\alpha_{wd}} \right) \left( \frac{v_{wd}}{\alpha_{wd}} \right)^{(\beta_{wd} - 1)} exp \left[ -\left( \frac{v_{wd}}{\alpha_{wd}} \right)^{\beta_{wd}} \right] 0 \leq v_{wd} < \infty \tag{24}$$

where $\alpha_{wd}$ and $\beta_{wd}$ denote the scale and shape parameters, respectively. The wind turbine's power output can be expressed as follows. [78]:

$$P_{wd}(v_{\omega d}) = \begin{cases} 0 & \text{for} & v_{\omega d} < v_{\omega i} \& v_{\omega} > v_{\omega o} \\ P_{wr}\left(\frac{v_{\omega d}-v_{\omega i}}{v_{\omega r}-v_{\omega i}}\right) & \text{for} & (v_{\omega i} \leq v_{\omega d} \leq v_{\omega r}) \\ P_{wr} & \text{for} & (v_{\omega r} < v_{wd} \leq v_{\omega o}) \end{cases}$$

(25)

where $P_{wr}$ denotes the wind turbine's rated power output while $v_{\omega i}$, $v_{\omega o}$, and $v_{\omega r}$ represent the wind turbine's cut-in, cut-out, and rated speeds, respectively. In this study, the wind farm comprises 25 turbines where the rated power of each turbine is 3 MW while the values of $v_{\omega r}$, $v_{\omega i}$ and $v_{\omega o}$ are 16, 3, and 25 m/s, respectively. For each scenario, the variability of wind speed can be determined using the following equation [44]:

$$\tau_{wind,scig} = \int_{V_{scig}^{min}}^{V_{scig}^{max}} f_{V_{wd}}(v_{wd})dv_{wd}$$

(26)

Here, $\tau_{wind,scig}$ refers to the probability of the wind falling within a specified scig-*th* scenario. Additionally, $v_{scig}^{min}$ and $v_{scig}^{max}$ are the boundaries of wind speed's interval at scig-*th* scenario, respectively.

### 3.3. Load uncertainty modeling

The uncertainty in loading was modeled using the normal PDF as $f_{Ld}(S_{Ld})$ which can be defined by the mean ($\mu_{Ld}$) and the standard deviation ($\sigma_{Ld}$) of the load [79]. The normal PDF is expressed as [41]:

$$f_{Ld}(S_{Ld}) = \frac{1}{\sqrt{2\pi}\sigma_{Ld}}exp\left[-\frac{(S_{Ld}-\mu_{Ld})2}{2\sigma_{Ld}^2}\right]$$

(27)

In this paper 800 scenarios have been generated using the Monte Carlo Simulation (MCS) method, random values are first generated using the Linear Congruential Generator (LCG). These values are then used as input variables in the inverse functions of the cumulative distribution functions (CDFs) for each probability density function (Weibull, lognormal, and normal). This technique is applied 800 times with independent random draws, generating 800 distinct scenarios. Each of these scenarios is defined as a unique combination of simulated random variables [80,81].

To optimize computational efficiency, the Scenario-based reduction (SBR) technique was employed, condensing the original 800 scenarios to just 15 representative ones [45,82,83]. The SBR follows structured steps that play a role in reducing the number of scenarios from 800 to 15. First, all scenarios are assigned based on uniform probability distribution, standing in for the variation in load demand, wind speed, and solar irradiance, adopting equal probabilities. Next, the similarity between the scenarios is assessed by calculating the distance between each pair within the set. Once the closest convergent scenarios have been identified, they are replaced by new scenarios representing their average. This process is then repeated until the required number of scenarios is reached, i.e., 15 scenarios.

It is worth mentioning here that selecting a high number of scenarios increases the computational burden while selecting a low number of scenarios, e.g., less than 10 reduces the accuracy of solutions. In this regard, 15 scenarios have been used to represent the uncertainty of the system.

In this study the parameters of the mean and the standard deviation of the irradiance are selected to be 5.5 and of 0.5, respectively. the wind parameters $\alpha_{wd}$ and $\beta_{wd}$ are selected to be 10.0434 and 2.5034, respectively. The load uncertainty parameters including $\sigma_{Ld}$ and $\mu_{Ld}$ are selected to be 10 and 70, respectively [39]. The wind speed, solar

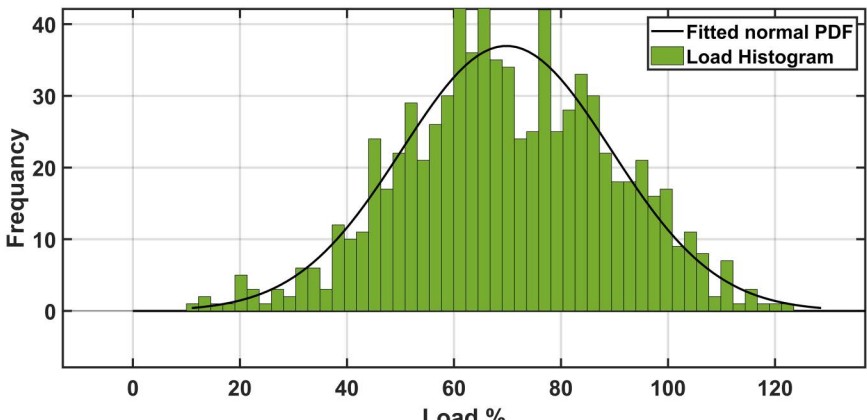

irradiation, and load demand scenario generation results are given in Figs 3–5 indicating the probabilities assigned to each scenario. These selected scenarios effectively capture the variability of irradiance, wind speed, and load, as shown in Table 2.

Figs 2–4 show the probabilities of power demand, wind speed, and solar irradiance which have been obtained by the MCS, respectively.

## 4. DO and MDO modeling

This section presents an in-depth analysis of the dandelion optimizer (DO) 's core structure and a comprehensive introduction to the improved version, MDO, developed and proposed in this study.

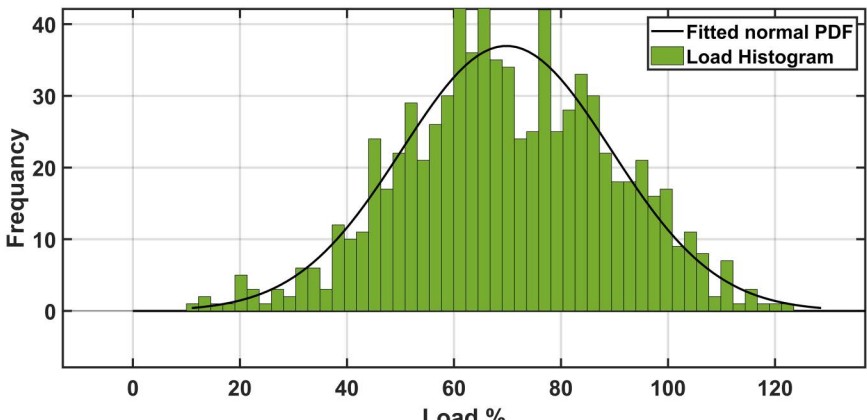

**Fig 2. The obtained scenarios of the load by the MCS.**

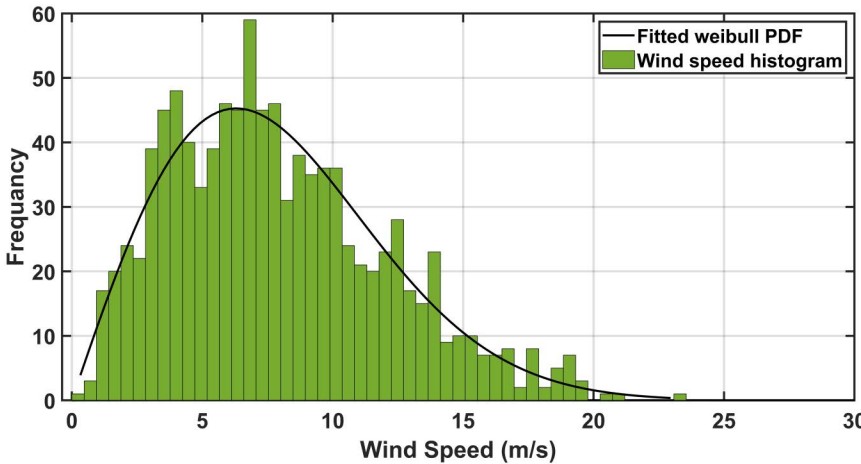

**Fig 3. The obtained scenarios of the wind speed by MCS.**

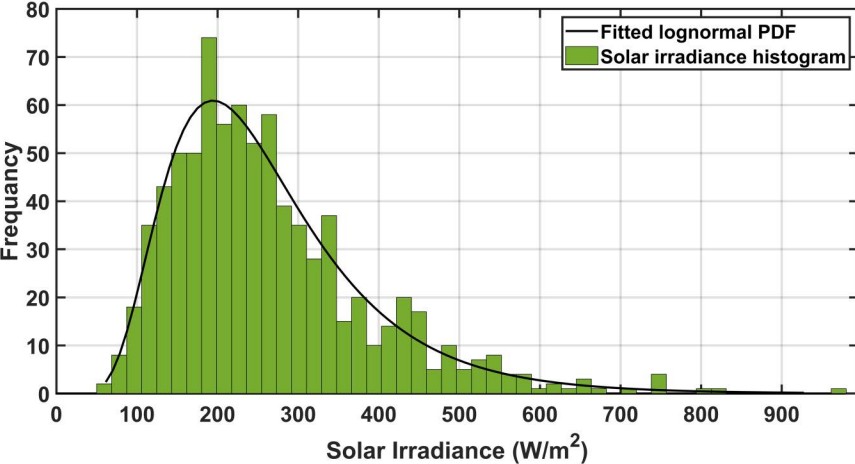

**Fig 4. The obtained scenarios of the solar irradiance by MCS.**

### 4.1. Overview of the dandelion optimizer (DO)

The Dandelion Optimizer (DO) represents an advanced optimization algorithm presented by Shijie Zhao et al. in 2022 [43]. The algorithm draws inspiration from the efficient dispersal mechanism of dandelion seeds. By leveraging wind dynamics, DO simulates the long-distance dispersal of seeds through four distinct phases, described as follows:

**4.1.1. Phase 1: Initialization.** In the DO approach, the initial population is initially generated arbitrarily inside the search zone. In each iteration, every particle in the swarm updates its position using the following equations:

$$X_i = \text{ rand } \times (UB - LB) + LB, i = 1, 2, \ldots, P \tag{28}$$

Here, $X_i$ denotes the solution candidate for the $i^{th}$ dandelion seeds, while $LB$ and $UB$ represent the lower and upper limits, respectively, as specified by the problem constraints. $P$ stands for the population size. The rand variable indicates a randomly generated value in the range [0, 1].

During initialization, the DO algorithm selects the individual with the least fitness value as the new candidate, regarded as the ideal location for dandelion seed growth. In contrast, the position and best solution of the most optimal particle are continuously stored in memory as $X_{new}$ and $f_{best}$.

**4.1.2. Phase 2: Rising.** Dandelion seeds fly through the air and sometimes scatter for miles depending on climatic changes such as wind speed, air humidity, etc. Consequently, the weather can be classified into the two following possibilities.

**Case 1: During clear weather**

Based on the wind speed, modeled using a lognormal distribution$\ln Y \sim N\left(\mu, \sigma^2\right)$, dandelions seeds can travel to distant regions on a clear day. In this case, the emphasis is on exploration. The vortex sucks up the dandelion seeds in a spiraling upward movement. With stronger wind, the seeds fly farther and spread higher. In this case, the relevant mathematical expression is:

$$X_{t+1} = X_t + \alpha \times v_x \times v_y \times \ln Y \times (X_s - X_t) \tag{29}$$

where $X_t$ indicates the dandelion seed's location throughout iteration t. $X_s$ denotes the randomly chosen location inside the search area in iteration $t$. Eq. (30) gives the equation for the randomly created position.

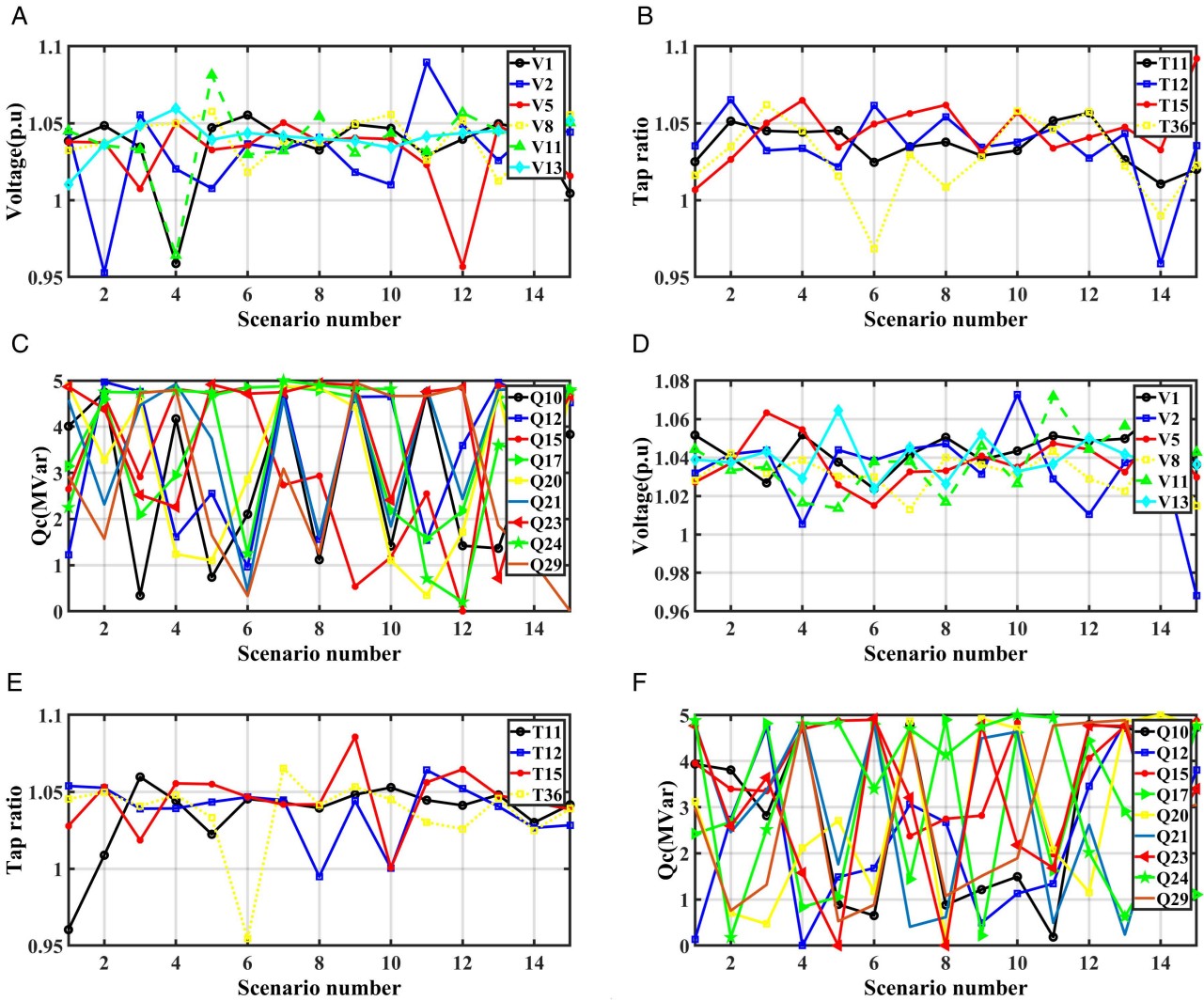

**Fig 5. Optimal system parameters for SEPL: (a) voltage, (b) tap ratio, (c) reactive power with RERs integration, and (d) voltage, (e) tap ratio, (f) reactive power without RERs integration.**

$$X_s = rand(1, Dim) \times (UB - LB) + LB \tag{30}$$

The mathematical formula for ln Y, which represents a lognormal distribution subject to $\mu = 0$ and $\sigma^2 = 1$ is presented as follows:

$$lnY = \begin{cases} \frac{1}{y\sqrt{2\pi}} exp\left[-\frac{1}{2\sigma^2}(ln\ y)^2\right] & y \geq 0 \\ 0 & y < 0 \end{cases} \tag{31}$$

$Y$ refers to the standard normal distribution N (0,1). α is an adaptive parameter between [0,1] used in order to guarantee a precise convergence following thorough global research and its mathematical expression is:

$$\alpha = rand() \times \left(\frac{1}{t_{max}^2}t^2 - \frac{2}{t_{max}}t + 1\right) \tag{32}$$

**Table 2. Reduced Scenarios for Load, Irradiance, Wind Speed, and Associated Probabilities.**

| Scenarios | Loading % | Irradiance (W/m2) | Wind speed (m/s) | Probability |
|---|---|---|---|---|
| 1 | 74.30 | 277.05 | 9.59 | 0.104 |
| 2 | 84.77 | 1289.85 | 9.51 | 0.001 |
| 3 | 72.59 | 1020.60 | 6.95 | 0.002 |
| 4 | 66.97 | 696.40 | 10.75 | 0.008 |
| 5 | 75.24 | 104.86 | 10.69 | 0.033 |
| 6 | 67.97 | 500.06 | 8.09 | 0.01 |
| 7 | 71.51 | 334.88 | 6.92 | 0.085 |
| 8 | 80.12 | 180.75 | 8.58 | 0.172 |
| 9 | 71.60 | 427.87 | 6.86 | 0.053 |
| 10 | 59.65 | 601.17 | 7.62 | 0.011 |
| 11 | 75.18 | 559.31 | 6.11 | 0.018 |
| 12 | 74.26 | 839.19 | 2.97 | 0.002 |
| 13 | 101.96 | 238.91 | 2.12 | 0.002 |
| 14 | 75.98 | 0.00 | 7.49 | 0.497 |
| 15 | 65.87 | 770.35 | 8.82 | 0.002 |

The lift coefficients of a dandelion, influenced by the vortex shedding effect, are expressed by $v_x$ and $v_y$. The force acting on the variable dimension is calculated using equation (33).

$$\begin{cases} r = \frac{1}{e^\theta} \\ v_x = r \times \cos\theta \\ v_y = r \times \sin\theta \end{cases}$$

(33)

Where $\theta$ represents a random number in the interval of [−π, π].

**Case 2: During rainy weather**

Due to factors like air resistance and humidity, dandelion seeds face challenges in ascending efficiently with the wind on rainy days. As a result, dandelion seeds are exploited in the surrounding areas, which corresponds to the mathematical expression below:

$$X_{t+1} = X_t * k$$

(34)

Here, $k$ governs the local search area of the dandelion and Eq. (35) is used to determine the search domain.

$$q = \frac{1}{t_{max}^2 - 2t_{max} + 1} t^2 - \frac{2}{t_{max}^2 - 2t_{max} + 1} t + 1 + \frac{1}{t_{max}^2 - 2t_{max} + 1}$$
$$k = 1 - rand() * q$$

(35)

Finally, the mathematical representation for dandelion seeds is:

$$X_{t+1} = \begin{cases} X_t + \alpha \times v_x \times v_y \times \ln Y \times (X_s - X_t) & \text{rand } n < 1.5 \\ X_t \times k & \text{else} \end{cases}$$

(36)

Where the random number produced by the function randn() adheres to the standard normal distribution (SND).

### 4.1.3. Phase 3: Descending.

During this phase, the DO algorithm emphasizes exploration as dandelion seeds adjust their direction in the global search space while descending steadily. Brownian motion, following a normal distribution, helps traverse diverse regions during iterative updates. After the rising stage, the average position reflects stability, guiding the population toward promising areas, as expressed mathematically.

$$X_{t+1} = X_t - \alpha \times \beta_t \times (X_{mean\_t} - \alpha \times \beta_t \times X_t) \tag{37}$$

Here, $\beta_t$ is a random variable representing Brownian motion, drawn from the SND. $X_{mean\_t}$ denotes the seed's mean position in the $i^{th}$ iteration and is expressed as:

$$X_{mean\_t} = \frac{1}{P} \sum_{i=1}^{P} X_i \tag{38}$$

### 4.1.4. Phase 4: Landing.

This phase of the Dandelion Optimizer (DO) focuses on exploitation, where seeds settle at random positions and the algorithm gradually converges toward the global optimum. The best solutions guide neighboring seeds to exploit promising areas more effectively. This ensures precise convergence to the optimal solution as iterations progress. This behavior is illustrated in the equation bellow.

$$X_{t+1} = X_{new} + levy(\lambda) \times \alpha \times (X_{new} - X_t \times \delta) \tag{39}$$

At the $i^{th}$ iteration, $X_{new}$ represents the dandelion seed's best placement. Indicating the Levy flight function, $levy(\lambda)$ is derived using Eq (40):

$$Levy(\lambda) = s \times \frac{z \times \sigma}{|h|^{\frac{1}{\beta}}} \tag{40}$$

Here, m is randomly selected from the range [0, 2], z and h are two random variables chosen from the interval [0, 1], and s is a constant value of 0.01. The mathematical expression for $\sigma$ is:

$$\sigma = \left( \frac{\Gamma(1+w) \times sin\left(\frac{\pi w}{2}\right)}{\Gamma\left(\frac{1+w}{2}\right) \times w \times 2^{\left(\frac{w-1}{2}\right)}} \right) \tag{41}$$

where $w$ is fixed at 1.5. δ is a linearly increasing function within the range [0, 2], and it is expressed as follows:

$$\delta = \frac{2z}{t_{max}} \tag{42}$$

## 4.2. Modified dandelion optimizer

The limitations of the original Dandelion Optimizer (DO) are that it prone to local optima and suffers from stagnation in case of solving the nonlinear objective functions like ORPD. In this regard, three modifications have been integrated for improving the searching ability of the DO using FDB, QOBL, and WFM. This subsection presents an enhanced version of the DO algorithm, termed MDO, which addresses the problem of stagnation in local optima and premature convergence. These occur due to a mismatch between exploitation and exploration and a lack of global exploration. DO addresses these issues with more effective strategies, including Fitness-Distance Balance (FDB), Quasi-Oppositional Based Learning (QOBL), and Weibull Flight Motion (WFM). It should be highlighted here that the WFM is utilized for boosting the

exploration of the applied optimizer and QOBL is used for boosting its exploitation process while the FDB is used for improving its global searching capability. These modifications can be described as follows:

**4.2.1. Quasi-oppositional-based-learning (QOBL).** Tizhoosh in [59] introduced the concept of Oppositional-Based Learning (OBL). This approach effectively enhances the convergence speed of various optimization algorithms by optimizing the trade-off between exploitation and exploration. Specifically, OBL is based on searching for the population opposite the current population, which may yield a better candidate solution. In this context, the quasi-opposite point, defined as the point located between the center and the opposite point, plays a crucial role in enabling a more balanced exploration of the solution domain. The QOBL was implemented to improve the searching ability of several optimization algorithms like teaching learning-based optimization [70], Lightning Attachment Procedure Optimizer [71], beluga whale optimizer [72], Runge-Kutta optimizer [38], Capuchin Search Algorithm [73], Dragonfly algorithm (DA) [74], differential evolution [75]. In our algorithm, opposite solutions are generated at each iteration by leveraging the best candidates in the population as a basis for opposition. The number of opposite solutions gradually decreases across iterations to expand the neighborhood and focus on exploiting the solution as it approaches optimality. However, this reduction in exploration may lead to premature convergence and an increased risk of being trapped in local optima, a phenomenon referred to as "capture." To mitigate this, opposite solutions are introduced to expand the search space and enhance exploration. This strategy is especially advantageous when the optimal solutions lie in directions that differ from those of the new candidates, as illustrated by the following equation.

$$x_i^o = UB_i + LB_i - x_i; x_i \in [UB, LB]; i = 1, 2, \ldots, Dim \tag{43}$$

The quasi-opposite point $x_i^{qo}$ is defined as the point between the center and the opposite point as follows:

$$x_i^{qo} = rand \left( \frac{UB_i + LB_i}{2}, UB_i + LB_i - x_i \right); i = 1, 2, \ldots, Dim \tag{44}$$

Where, *UB* and *LB* are the upper and the lower limits of the control variables. *Dim* denotes the dimension of the studied problem.

**4.2.2. The fitness-distance balance (FDB).** The FDB method, developed by Kahraman et al [58]. It was applied with several algorithms like fractal search algorithm [63], artificial ecosystem optimization [64], differential evolution [64], hybrid particle swarm optimizer [65], coyote optimization algorithm [66], teaching–learning-based optimization algorithm [67], Supply-demand-based optimization [68], Runge–Kutta algorithm [69]. The FDB method is a powerful selection approach aimed at efficiently directing the metaheuristic optimization process. Its selection strategy is based on two key factors: the fitness evaluation of the potential solutions. A lower fitness value signifies a higher-quality solution, aiding in effectively directing the search. The second key parameter is the separation between a candidate solution and the optimal solution (X*best*). A greater gap between two candidates (X*i* and X*best*) within the search space is beneficial, as it indicates diversity and the potential for complementing weaknesses. The FDB method determines the score of each candidate in the population by incorporating both fitness and distance metrics. When a solution with a favorable fitness value also exhibits a significant separation, it strongly suggests that it will contribute meaningfully to the search process. The step-by-step procedure for selecting candidates within the FDB framework is described below.

The distance and fitness vectors calculation: The first task is to compute the distance between each candidate X*i* and the best solution X*best* in the population. This distance is calculated using the Euclidean formula, as shown in Eq. (45), where i = 1, 2, …, P:

$$DG_i = \sqrt{(x_{i1} - x_{best1})^2 + (x_{i2} - x_{best2})^2 + \cdots + (x_{i,d} - x_{bestDim]})^2} \tag{45}$$

After the distances are computed, a vector of these distance values is represented as follows. This vector represents the relative distances of all solutions to the best solution:

$$DG = [DG_1, DG_2, \cdots, DG_P] \tag{46}$$

In parallel, the fitness values for each solution are also stored in a fitness vector, representing the quality of each solution. The fitness vector is structured as:

$$f = [f_1, f_2, \cdots, f_P] \tag{47}$$

FDB score generation: Each solution's score is determined based on the normalized distance and fitness values, which are computed as follows.

$$norm\, DG_i = \frac{DG_i - min(DG)}{max(DG) - min(DG)} \tag{48}$$

$$norm\, f_i = \frac{f_i - min(f)}{max(f) - min(f)} \tag{49}$$

Then, the scores of particles are calculated as follows:

$$Sb_i = \alpha \times (1 - norm\, f_i) + (1 - \alpha) \times\ norm\, DG_i \tag{50}$$

In which

$$\alpha = 0.5 \times \left(1 + \frac{t}{t_{max}}\right) \tag{51}$$

where $t_{max}$ and $t$ are the maximum iteration number and $t$ is the current iteration.

The Sb vector, as outlined in Eq. (52), contains the FDB score values for the potential solutions. These solutions are selected according to the Sb-vector to guide the search process of metaheuristic algorithms effectively.

$$Sb = [S_1, S_2, \cdots, S_P] \tag{52}$$

Further details regarding the FDB method are available in the referenced study [58].

**4.2.3. The Weibull flight motion strategy (WFM).** The third modified strategy involves Weibull flight motion, derived from the Weibull flight distribution. This approach aims to mitigate premature convergence to local optima and enhance the diversity of the population. By incorporating the Weibull flight motion, the algorithm enhances its global search capabilities, helping to avoid local optima and accelerate convergence. The Weibull flight distribution can be formulated as follows [60]:

$$x_{new,i} = x_{new,i} + Step \tag{53}$$

In which

$$Step =\ wbr\,(1, 1, [1, Dim]) \cdot \times sign(rand(1, Dim) - 0.5) \tag{54}$$

Where *wbr* represents a random drawn from the Weibull distribution. The Sign () function returns a vector of values –1 and +1.

It should be mentioned that the proposed MDO has a high level of computational complexity, because, as mentioned earlier, DO is based on three stages, including rising, landing, and descending. Hence, the computational complexity of the DO is $O(M \times T \times pop \times Dim \times fit)$, where M, T, pop, Dim, and fit are the current optimal solution, the maximum iteration number, the number of populations, the problem dimension, and the objective function [84]. The MDO is based on three modifications, including the WFM, QOBL, and FDB, Consequently, the computational complexity of the MDO is $O(M \times T \times pop \times Dim \times fit + 3(M \times T \times pop \times Dim \times fit))$. Hence, the computational complexity of the MDO is $O(4(M \times T \times pop \times Dim \times fit))$.

### 4.2.4. Pseudo-code of DO.

```
Input: Initialize parameters: P, Dim, tmax, UB, LB.
Output: The best solution X_best and its corresponding fitness value f_best.
Randomly initialize the population of dandelion seeds X_i using Eq. (20)
1. Evaluate the fitness of each seed and set X_best as the seed with the best fitness
2. while (t ≤ t_max) do
3. for i=1 to P do
                        /*Rising stage*/
4.   if randn () < 1.5 do    % Clear weather
5.     Udjust the seed position based on the Eq. (29)
6.   else if % Rainy weather
7.     Udjust the seed position based on the Eq. (34)
8.   end if
                        /*Descending phase*/
9.   Dandelion seeds descend using Brownian motion by Eq. (37)
                        /*Landing phase*/
10.    Seed settles using a Levy flight for local exploitation by Eq. (39)
11.    Update X_new according to fitness values
12.    if X_new < X_best do
13.    X_best = X_new and f_best = f (X_new)
14.    end if
15. end for
                        /*Update dandelion seeds based on QOBL*/
16. for i=1 to P do
17.    Update X_new based on Eq. (43)
18. if X_new < X_best do
19.    X_best = X_new and f_best = f (X_new)
20.    end if
21. end for
                        /*Update dandelion seeds based on FDB*/
22. for i=1 to P do
23.    Determine the distance between X_i and X_new based on Eq. (45)
24.    Generate the distance vector (DP) as defined in Eq. (46)
25.    end for
26. for i=1 to P do
27.    Compute the score value of each solution using Eq. (50)
28.    Create the FDB-based score (SP) as shown in Eq. (52)
29. end for
30. Select candidates for the search process by their scores (SP).
31.    if X_new < X_best do
32.       X_best = X_new and f_best = f (X_new)
33.    end if
                        /*Update dandelion seeds based on WFM*/
34. for i=1 to P do
35.    Update X_new based Eq. (53)
```

```
36.      if  X_new < X_best  do
37.         X_best = X_new  and  f_best = f  (X_new)
38.      end if
39.  end for
40.      Increment t by 1
41.      end while
42.  Return:  X_best
```

## 5. Analysis and discussion of simulation results

To demonstrate the superior capability of the MDO algorithm in addressing the SORPD problem, we performed extensive simulations on two case studies based on the IEEE 30-bus network. These simulations were carefully structured to evaluate performance under conditions with and without the integration of RERs uncertainties. The simulations were conducted on an Intel Core i7 PC, 1.80 GHz, 8 GB of RAM, and using MATLAB R2019b.

### 5.1. Simulation configuration

The studied system is the IEEE 30-bus transmission system that consists of 30 buses, 41 branches, 6 thermal generators, 9 shunt VAR compensators, and 4 tap-changing transformers. The load demands are 126.2 MVAr for reactive power and 283.2 MW for active power, with generator bus voltages and transformer tap settings ranging from 0.9 to 1.1 p.u., while shunt VAR compensators were regulated within the 0–5 MVAr range.

In this system, a wind farm is integrated which consists of 25 turbines, each with a 3 MW rating as well as a solar photovoltaic (PV) unit with a rated capacity of 50 MW. The selected turbine's wind speed characteristics include the cut-in speed, cut-out speed, and the cut-out speed, the rated wind speed are 3 m/s, 25 m/s, and 16 m/s, respectively. The wind speed follows a Weibull distribution with shape and scale factors of 2.5034 and 10.0434, respectively [39]. The mean and the standard deviation of the load demand are 70 and 10, respectively [85].

The MDO, DO, SCSO, GTO, HS, and BWO algorithms are applied to solve the SORPD problem. The parameters of these algorithms are outlined in Table 3. To ensure a fair comparison, all algorithms were evaluated using 25 search agents and 100 maximum iterations. The evaluation was conducted over 25 independent runs across 15 scenarios. The effectiveness of these algorithms in solving the SORPD problem was demonstrated by optimizing the objective functions

Table 3. Description of algorithm parameters.

| Algorithm | Parameter | Value |
|---|---|---|
| SCSO [86] | Sensitivity range (rg) | [2,0] |
|  | Phases control range (R) | [− 2rg, 2rg] |
| GTO [87] | β | 3 |
|  | p | 0.03 |
|  | w | 0.8 |
| HS [88] | HMCR | 0.95 |
|  | PAR | 0.45 |
| BWO [89] | Wf | [0.1 0.05] |
| DO [90] | α | [0,1] |
|  | k | [0,1] |
| MDO | α | [0,1] |
|  | k | [0,1] |
|  | sign | [−1,1] |

including *SEPL* and *SEVS* with and without the inclusion of RERs. Initially, the SORPD problem was addressed without considering RERs, and subsequently, it was re-evaluated after incorporating RERs.

The detailed results of the analyzed cases are presented below.

## 5.2. Results discussion

### 5.2.1. Solving the SORPD for the *SEPL*.
In this section the MDO is applied for SORPD solution for *SEPL* with and without considering RERs. Table 4 lists the simulation results for *SEPL*, comparing outcomes with and without the inclusion of uncertainties in RERs. The SORPD is solved under the 15 generated scenarios. Table 4 lists the active power

**Table 4. Simulation results for *SEPL* with and without RERs uncertainties.**

| Scenario No. | $P_{PV}$(MW) | $P_{Wind}$(MW) | $P_{Loss}$(MW) | EPL(MW) | VS (p.u) | EVS (p.u) |
|---|---|---|---|---|---|---|
| **Without integration of RERs** | | | | | | |
| 1 | 0 | 0 | 1.80204 | 0.18741 | 0.10368 | 0.01078 |
| 2 | 0 | 0 | 2.98553 | 0.00298 | 0.12784 | 0.00012 |
| 3 | 0 | 0 | 1.94966 | 0.00389 | 0.10594 | 0.00021 |
| 4 | 0 | 0 | 1.31242 | 0.01049 | 0.09328 | 0.00074 |
| 5 | 0 | 0 | 1.99130 | 0.06571 | 0.1117 | 0.00368 |
| 6 | 0 | 0 | 1.58545 | 0.01585 | 0.09165 | 0.00091 |
| 7 | 0 | 0 | 1.67828 | 0.14265 | 0.10262 | 0.00872 |
| 8 | 0 | 0 | 2.31996 | 0.39903 | 0.11964 | 0.02057 |
| 9 | 0 | 0 | 1.60991 | 0.08532 | 0.10402 | 0.00551 |
| 10 | 0 | 0 | 1.06505 | 0.01171 | 0.08506 | 0.00093 |
| 11 | 0 | 0 | 1.87289 | 0.03371 | 0.10405 | 0.00187 |
| 12 | 0 | 0 | 1.82994 | 0.00366 | 0.10174 | 0.00020 |
| 13 | 0 | 0 | 5.65432 | 0.01130 | 0.14552 | 0.00029 |
| 14 | 0 | 0 | 1.81656 | 0.90283 | 0.10314 | 0.05126 |
| 15 | 0 | 0 | 1.58223 | 0.00316 | 0.09341 | 0.00018 |
| ∑ | | | | **1.8798** | | 0.1060 |
| **With integration of RERs** | | | | | | |
| 1 | 11.35921 | 35.4965 | 1.04252 | 0.10842 | 0.09542 | 0.00992 |
| 2 | 52.88386 | 35.04147 | 4.36270 | 0.00436 | 0.10583 | 0.00010 |
| 3 | 41.84453 | 21.27721 | 2.16105 | 0.00432 | 0.087174 | 0.00017 |
| 4 | 28.55252 | 41.71422 | 2.76717 | 0.02213 | 0.08492 | 0.00067 |
| 5 | 3.75650 | 41.38849 | 1.18314 | 0.03904 | 0.09771 | 0.00322 |
| 6 | 20.50227 | 27.42668 | 1.53032 | 0.01530 | 0.08888 | 0.00088 |
| 7 | 13.73016 | 21.09412 | 1.042977 | 0.08865 | 0.08995 | 0.00764 |
| 8 | 11.35921 | 35.4965 | 1.04252 | 0.10842 | 0.09542 | 0.00992 |
| 9 | 52.88386 | 35.04147 | 4.36270 | 0.00436 | 0.10583 | 0.00010 |
| 10 | 41.84453 | 21.27721 | 2.16105 | 0.00432 | 0.08717 | 0.00017 |
| 11 | 28.55252 | 41.71422 | 2.76717 | 0.02213 | 0.08492 | 0.00067 |
| 12 | 3.75650 | 41.38849 | 1.18314 | 0.03904 | 0.09771 | 0.00322 |
| 13 | 20.50227 | 27.42668 | 1.53032 | 0.01530 | 0.08888 | 0.00088 |
| 14 | 13.73016 | 21.09412 | 1.04297 | 0.08865 | 0.08995 | 0.00764 |
| 15 | 11.35921 | 35.4965 | 1.04252 | 0.10842 | 0.09542 | 0.00992 |
| ∑ | | | | **1.2628** | | 0.0985 |

loss, the voltage stability, the expected power loss and the expected voltage stability, the generated powers of WTs and PV system for each scenario. The *SEPL* decreased from 1.8798 MW without RERs to 1.2628 MW with RERs or the improvement in reduction summation of the expected power losses is about 32.82%. These results not only validate the effectiveness of the MDO approach but also underscore the crucial role of integrating RER uncertainties in power loss optimization.

Fig 5(a), 5(b), and 5(c) show the optimal values of the generator voltages, transformers taps and the reactive powers of capacitor banks for *SEPL* without RERs while Fig 5(d), 5(e), and 5(f) show the optimal values of these paramters with RERs, respectively.

The optimal location and size of the PV system are at bus 20 and 41 MW while optimal location and size of the WTs system are at bus 7 and 70 MW, respectively. Fig 6 shows the output power of the RERs with and without RERs for this case at each scenario.

**5.2.2. Solving the SORPD for the for the SEVS.** In this section the MDO is applied for SORPD solution for *SEVS* with and without considering RERs. Table 5 lists the optimal setting of the system components across 15 different scenarios for improving SEVS with and without RERs. Table 5 provides a summary of the simulation results across 15 different scenarios. Table 5 lists the active power loss, the voltage stability, the expected power loss and the expected voltage stability, the generated powers of WTs and PV system of each scenario for *SEVS*. The value of SEVS without inclusion RERs is 0.0944 p.u while with inclusion the RERs, the value of *SEVS* with RERs is 0.0617 p.u. or the system stability is enhanced by 34.64% with inclusion of the RERs.

Fig 7(a), 7(b), and 7(c) show the optimal values of the generator voltages, transformers taps and the reactive powers of capacitor banks for *SEVS* without RERs while Fig 7(d), 7(e), and 7(f) show the optimal values of these paramters with RERs, respectively. In this case, the optimal location and size of the PV system are at bus 29 and 42 MW while optimal location and size of the WTs system are at bus 20 and 34 MW, respectively. Fig 8 shows the output power of the RERs with and without RERs for this case at each scenario.

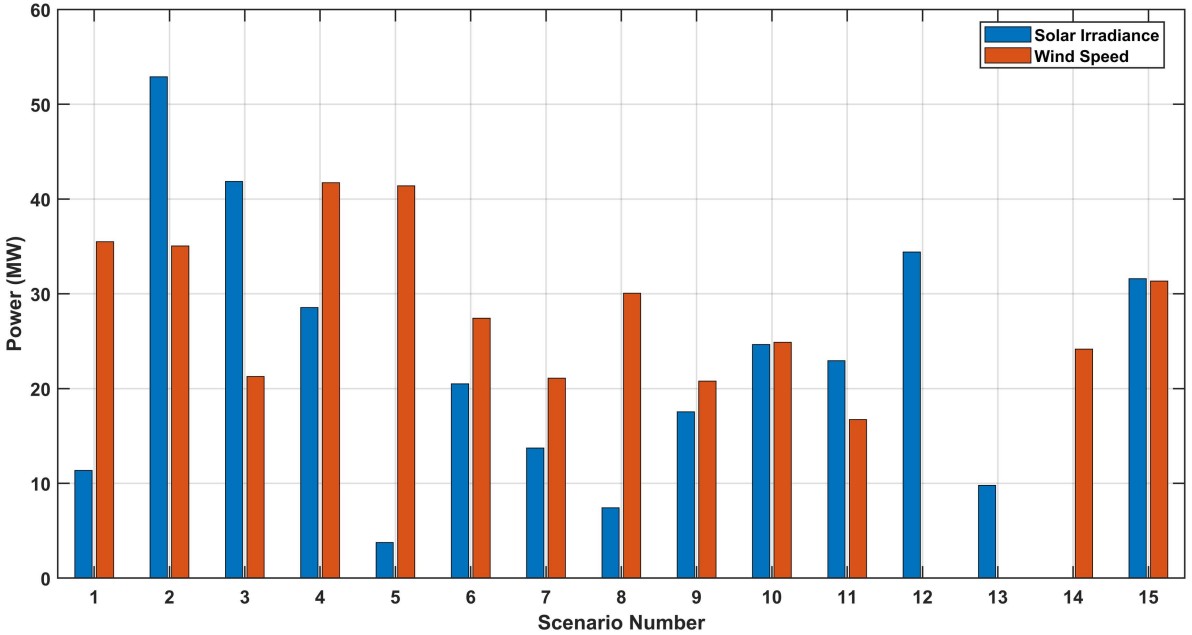

**Fig 6. The power generated for the SEPL.**

**Table 5. Simulation Results for *SEVS* with and without RERs uncertainties.**

| Scenario No. | $P_{PV}$(MW) | $P_{Wind}$(MW) | $P_{Loss}$(MW) | EPL(MW) | VS (p.u) | EVS (p.u) |
|---|---|---|---|---|---|---|
| **Without the integration of RERs** | | | | | | |
| 1 | 0 | 0 | 2.37667 | 0.24717 | 0.09672 | 0.01006 |
| 2 | 0 | 0 | 3.33841 | 0.00333 | 0.12070 | 0.00012 |
| 3 | 0 | 0 | 3.85060 | 0.00770 | 0.0956 | 0.00019 |
| 4 | 0 | 0 | 1.78102 | 0.01424 | 0.09681 | 0.00077 |
| 5 | 0 | 0 | 2.33146 | 0.07693 | 0.09747 | 0.00321 |
| 6 | 0 | 0 | 1.99218 | 0.01992 | 0.08514 | 0.00085 |
| 7 | 0 | 0 | 2.53305 | 0.2153 | 0.09045 | 0.00768 |
| 8 | 0 | 0 | 2.76189 | 0.47504 | 0.09995 | 0.01719 |
| 9 | 0 | 0 | 2.72594 | 0.14447 | 0.09795 | 0.00519 |
| 10 | 0 | 0 | 1.79234 | 0.01971 | 0.08575 | 0.00094 |
| 11 | 0 | 0 | 2.77892 | 0.05002 | 0.10237 | 0.00184 |
| 12 | 0 | 0 | 2.32256 | 0.00464 | 0.10555 | 0.00021 |
| 13 | 0 | 0 | 5.54874 | 0.01109 | 0.13787 | 0.00027 |
| 14 | 0 | 0 | 2.12645 | 1.05684 | 0.09195 | 0.04570 |
| 15 | 0 | 0 | 1.95636 | 0.00391 | 0.09217 | 0.00018 |
| ∑ | | | | 2.3504 | | 0.0944 |
| **With the integration of RERs** | | | | | | |
| 1 | 11.91331 | 21.2979 | 1.65494 | 0.17211 | 0.05156 | 0.00536 |
| 2 | 55.46356 | 21.02488 | 4.96484 | 0.00496 | 0.05835 | 5.84E-05 |
| 3 | 43.88572 | 12.76633 | 5.52771 | 0.01105 | 0.04799 | 9.6E-05 |
| 4 | 29.94533 | 25.02853 | 2.97735 | 0.02381 | 0.08343 | 0.00066 |
| 5 | 3.93974 | 24.83309 | 2.22318 | 0.07336 | 0.06866 | 0.00226 |
| 6 | 21.50238 | 16.45601 | 1.63704 | 0.01637 | 0.05537 | 0.00055 |
| 7 | 14.39992 | 12.65647 | 1.34531 | 0.11435 | 0.04941 | 0.00420 |
| 8 | 7.77227 | 18.034 | 1.966054 | 0.33816 | 0.05519 | 0.00949 |
| 9 | 18.39844 | 12.4696 | 2.79795 | 0.14829 | 0.05537 | 0.00293 |
| 10 | 25.85024 | 14.92205 | 2.83117 | 0.03114 | 0.04072 | 0.00044 |
| 11 | 24.05019 | 10.03432 | 1.42499 | 0.02565 | 0.06107 | 0.00109 |
| 12 | 36.08525 | 0 | 3.16112 | 0.00632 | 0.08685 | 0.00017 |
| 13 | 10.27299 | 0 | 5.48760 | 0.01097 | 0.14109 | 0.00028 |
| 14 | 0 | 14.49452 | 2.33872 | 1.16234 | 0.0682 | 0.03394 |
| 15 | 33.12497 | 18.8022 | 2.654977 | 0.00531 | 0.04993 | 9.99E-05 |
| ∑ | | | | 2.1442 | | 0.0617 |

**5.2.3. Convergence behavior analysis.** The analysis of the convergence of summation of expected power losses (SEPL) was conducted using various optimization methods: Modified Differential Operator (MDO), conventional Differential Operator (DO), Sine Cosine Swarm Optimization (SCSO), Genetic Turbine Optimizer (GTO), Beluga whale optimization (BWO), and Harmony Search (HS). The study is carried out across 100 iterations and examined two scenarios: with and without the inclusion of RERs. Fig 9 shows the convergence of the proposed algorithm and the other comparative algorithms for SEPL with and without RERs. The results showed that MDO consistently has the best convergence characteristics for the considered objective function compared to other techniques.

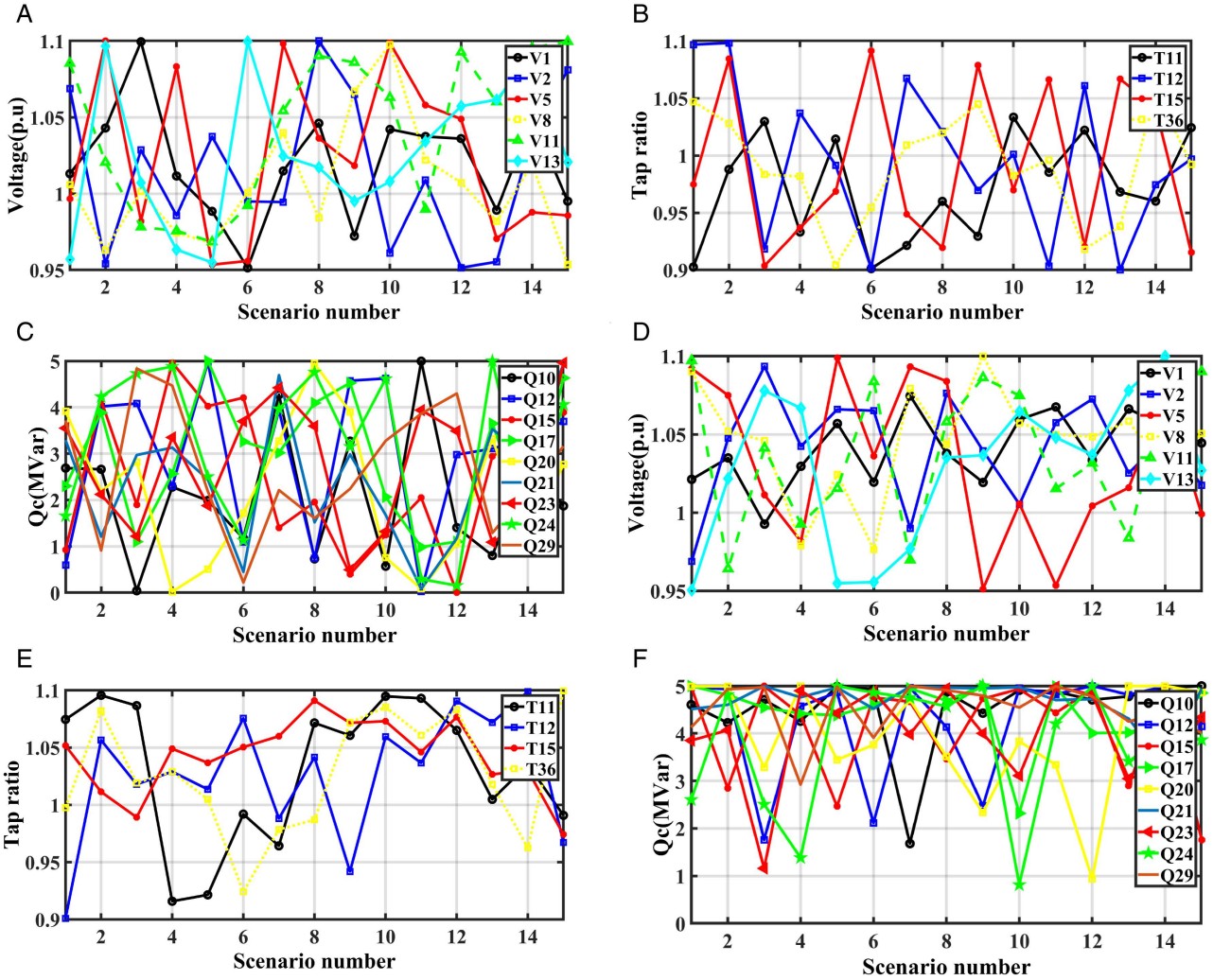

**Fig 7. Optimal system parameters for SEVS: (a) voltage, (b) tap Ratio, (c) reactive power with RERs integration, and (d) voltage, (e) tap ratio, (f) reactive power without RERs integration.**

Additionally, Fig 10 presents the convergence curves for the summation of expected voltage stability (SEVS) with and without inclusion RERs. According to Fig 10, the MDO has the best convergence performance compared with other algorithms.

**5.2.4. Voltage profiles analysis.** Optimization of SEPL, and SEVS parameters using MDO enabled can improve the voltage profiles for generated scenarios with and without the integration of RERs, as shown in Fig 11. The analysis indicates that RERs integration can enhance the voltage profile of the IEEE 30-bus network, confirming the effectiveness of MDO in improving overall stability and performance.

**5.2.5. Box plot analysis.** Boxplots serve as powerful visual tools for depicting data distribution across quartiles. In a boxplot, the ends represent the minimum and maximum values, indicating the lower and upper bounds of the whiskers, the interquartile range, spanning the first and third quartiles, is delineated by the central box. A narrower box indicates greater consensus and higher consistency within the dataset. Fig 12 displays the boxplots of the obtained results by the proposed MDO and the other optimization techniques for SEPL and SEVS with and without integration of RERs.

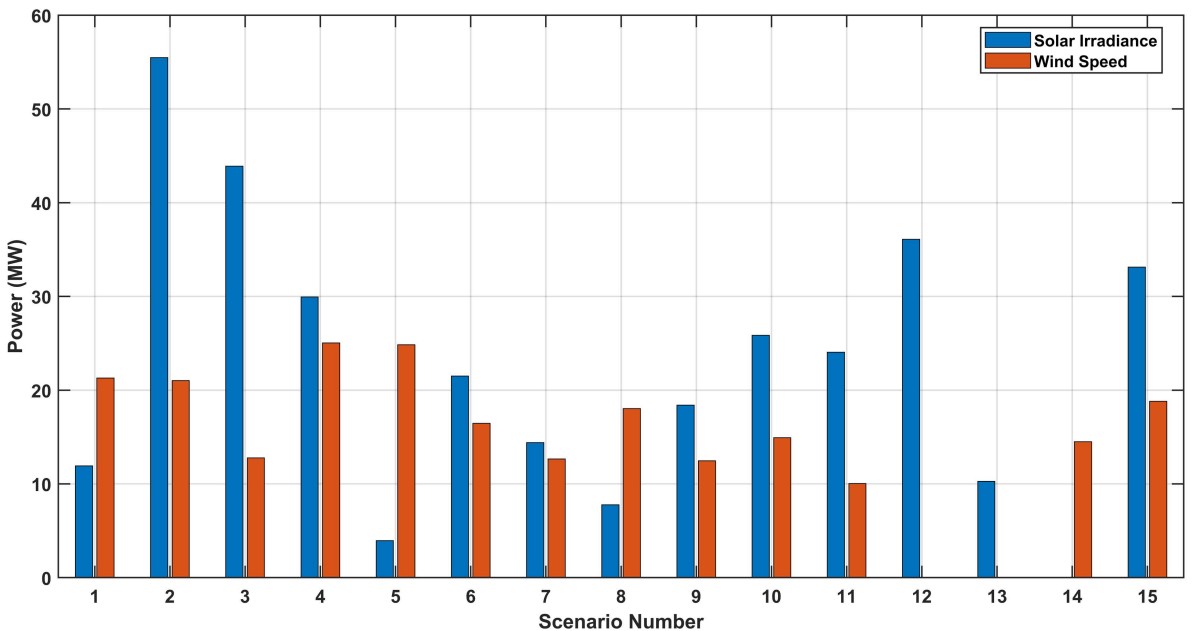

**Fig 8. The power generated for the SEVS.**

According to Fig 12, the MDO method consistently achieved the most favorable distributions, recording the lowest values for SEPL, and SEVS in both cases, outperforming all other methods. In contrast, it can be observed that the BWO method results in a large discrepancy between the minimum and maximum values, indicating stress violations during the test. This occurs because BWO distributes its candidate solutions widely across the search space to avoid premature convergence and to thoroughly explore potential optimal solutions. While this intensive exploration helps avoid local minima, it can also lead to less consistent or higher objective function values in some runs. It is important to note that Fig 12 shows the distribution of the objective function, which incorporates both the objective value and system constraints.

**5.2.6. Sensitivity analysis.** The sensitivity analyses of SEPL and SEVS are presented in Tables 6 and 7, respectively. The results are unambiguous and clearly demonstrate robustness. Of the proposed MDO approach as the number of scenarios increases. Unlike Algorithms used in this study which experience fluctuations and reduced performance due to variations in the number of scenarios, the MDO systematically achieves the lowest values. This is a clear demonstration of performance stability and reliability. This gradual improvement as scenarios increase highlights MDO's ability to effectively manage complexity and uncertainty, reinforcing its relevance for practical applications.

**5.2.7. Statistical analysis.** Table 8 presents statistical comparisons of SEPL and SEVS across different algorithms with and without the integration of RERs. The results show that the MDO algorithm consistently outperforms other methods such as SCSO, HS, GTO, BWO, and traditional DO, in optimizing SEPL and SEVS, both cases. In the case of RERs integration, MDO clearly stands out from other algorithms in terms of SEPL minimization and SEVS improvement. It achieves a very low mean of 9.87609 MW and a best solution of 1.26277 MW for the SEPL objective, with a narrow confidence interval of [0, 21.32], a sign of remarkable stability. For the SEVS objective, although the average is slightly higher (24.08707 p.u), the best solution obtained (0.06167 p.u) remains the best performance of all the algorithms. This shows that, when considering the uncertainty associated with renewable energy sources, MDO combines efficiency, consistency, and optimal solution quality, with precise confidence intervals [6.10, 42.08] confirming its consistency. In the case without the integration of RERs, MDO confirms its superiority, particularly in terms of the best solutions found. For

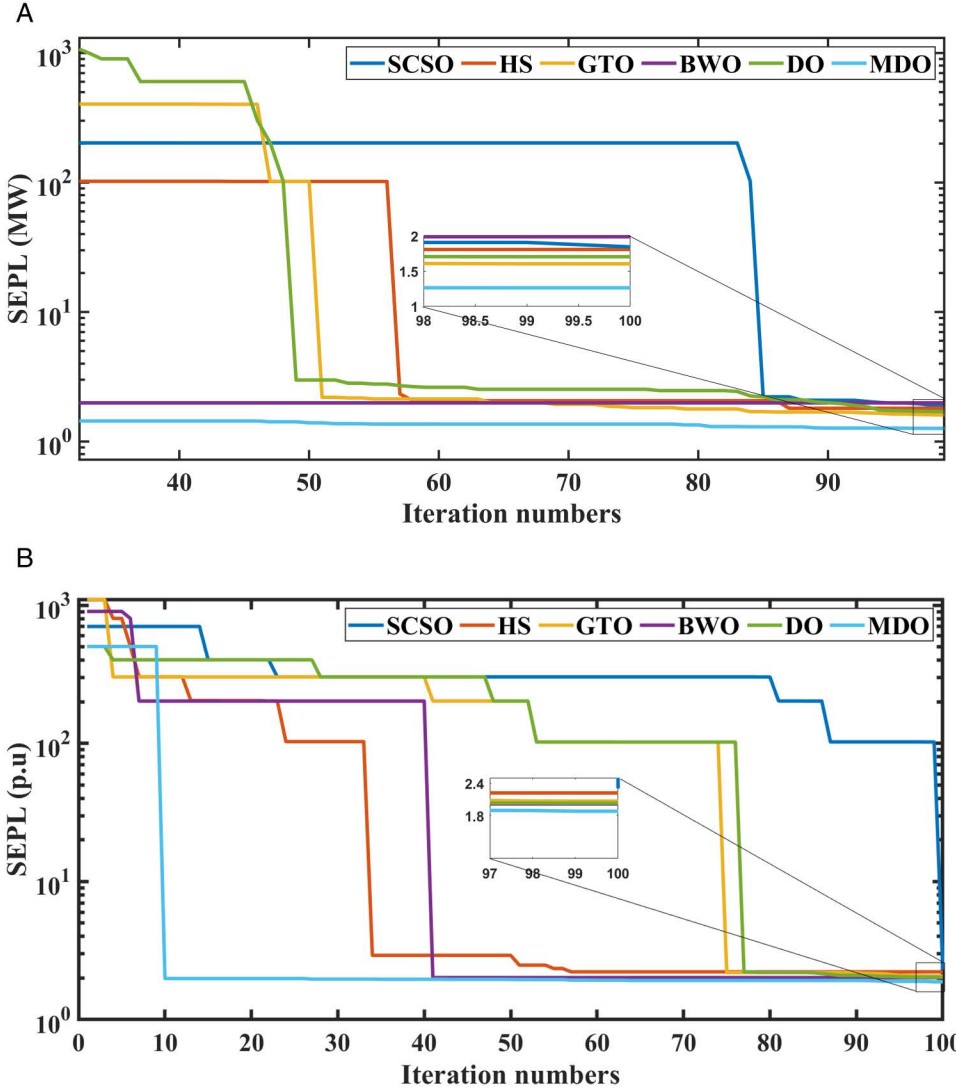

**Fig 9. Convergence curves of MDO and other methods for SEPL, (a) with integration of RERs and (b) without integration of RERs.**

SEPL, it achieves 1.87977 MW at best, a result that remains superior to those of the other algorithms tested. For SEVS, it also retained the best performance with 0.09444 p.u. These results demonstrate that MDO remains the best-performing and most reliable algorithm for identifying high-quality solutions, with reasonable confidence intervals: [2.703, 33.64] for SEPL and [0, 19.51] for SEVS, thus supporting its high level of robustness, whatever the execution cases (with and without RERs integration).

In summary, MDO consistently achieves the highest average ranking, demonstrating more favorable average values, lower standard deviations, and better-controlled maximum and minimum values, particularly when renewable energy is incorporated. The MDO's superiority is due to the applied improvement strategies, including the WFM, the QOBL and the FDB, which boost the searching capabilities of the MDO compared to other techniques.

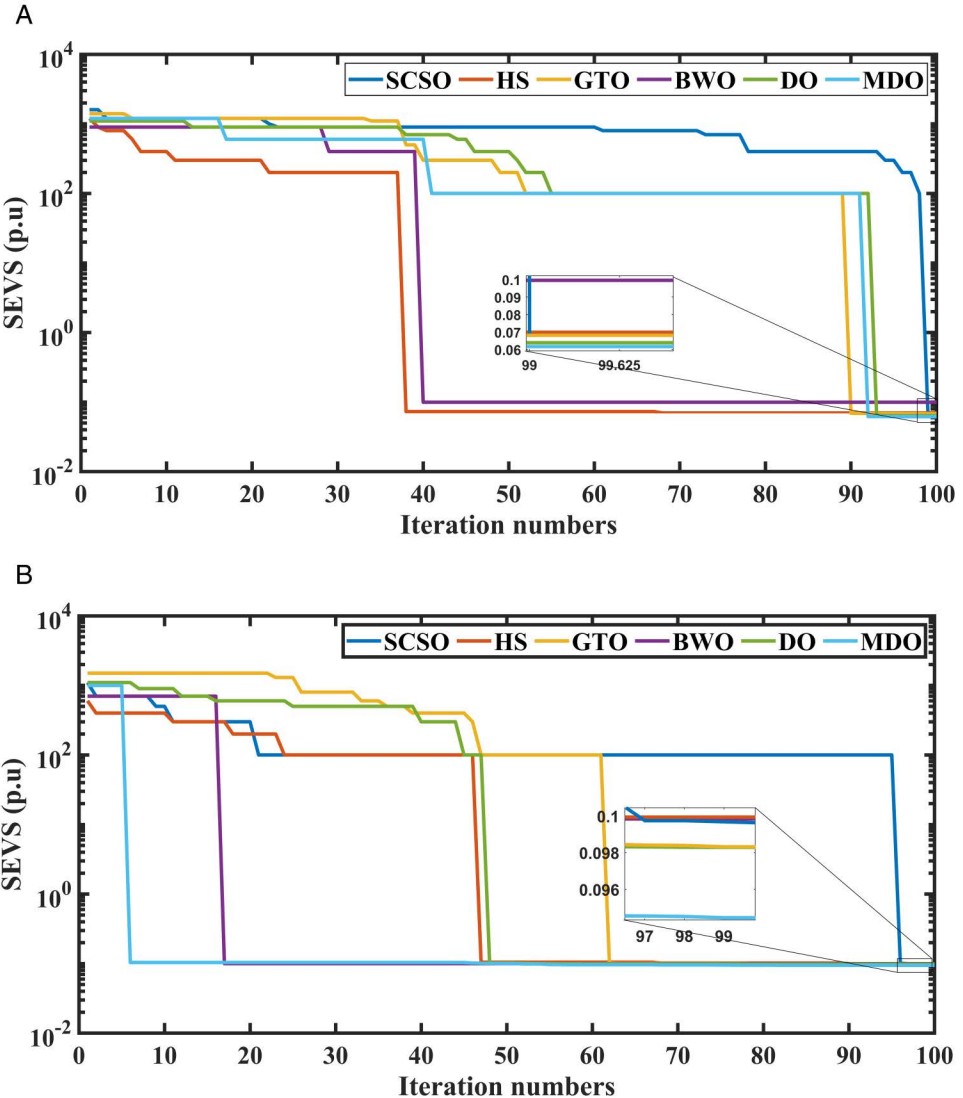

**Fig 10. Convergence curves of MDO and other optimization methods for SEVS, (a) with integration of RERs and (b) without integration of RERs.**

## 6. Conclusion

In this study the Stochastic Optimal Reactive Power Dispatch (SORPD) was solved optimally for the IEEE 30-bus. The SORPD has been solved considering the summations of expected power losses (SEPL), and the expected voltage stability (SEVS). The problem has been solved with and without the inclusion of the RERs to realize the effects of incorporating the RERs in the system performance. The SORPD has been solved using a Modified Dandelion optimizer MDO that is based on the integration of three modifications including QOBL, FDB and WFM. The uncertainties of demand and the yielded powers of the RERs have been represented in terms of the load demand, the wind speed and the irradiance using Weibull PDF, Lognormal BDF, and normal BDF while MCS methods were used to create 800 scenarios of the uncertain parameters, which have been decreased to 15 scenarios. The ORPD was solved for each scenario in which the voltages of the generation bus, the taps of the transformers and the reactive powers have been assigned optimally for the 15

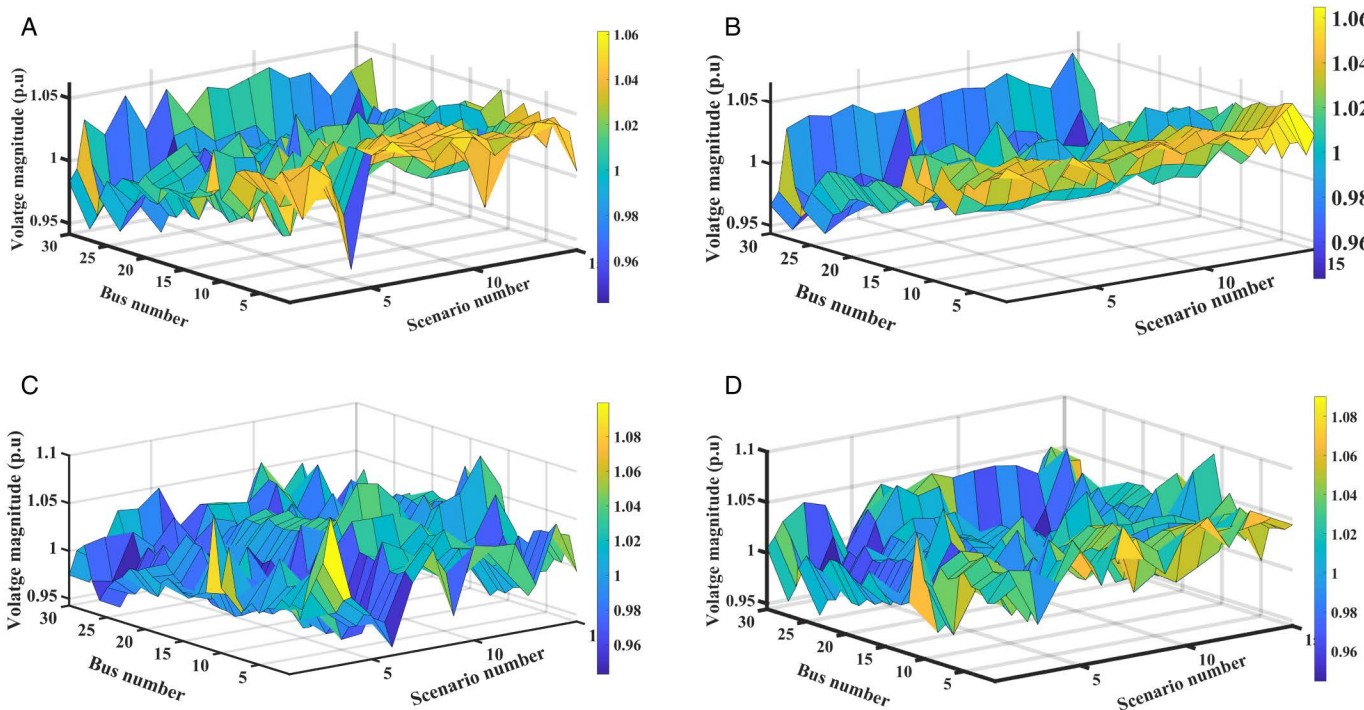

**Fig 11. The system voltage profile: (a) SEPL with RERs, (b) SEPL without RERs, (c) SEVS with RERs, and (d) SEVS without RERs.**

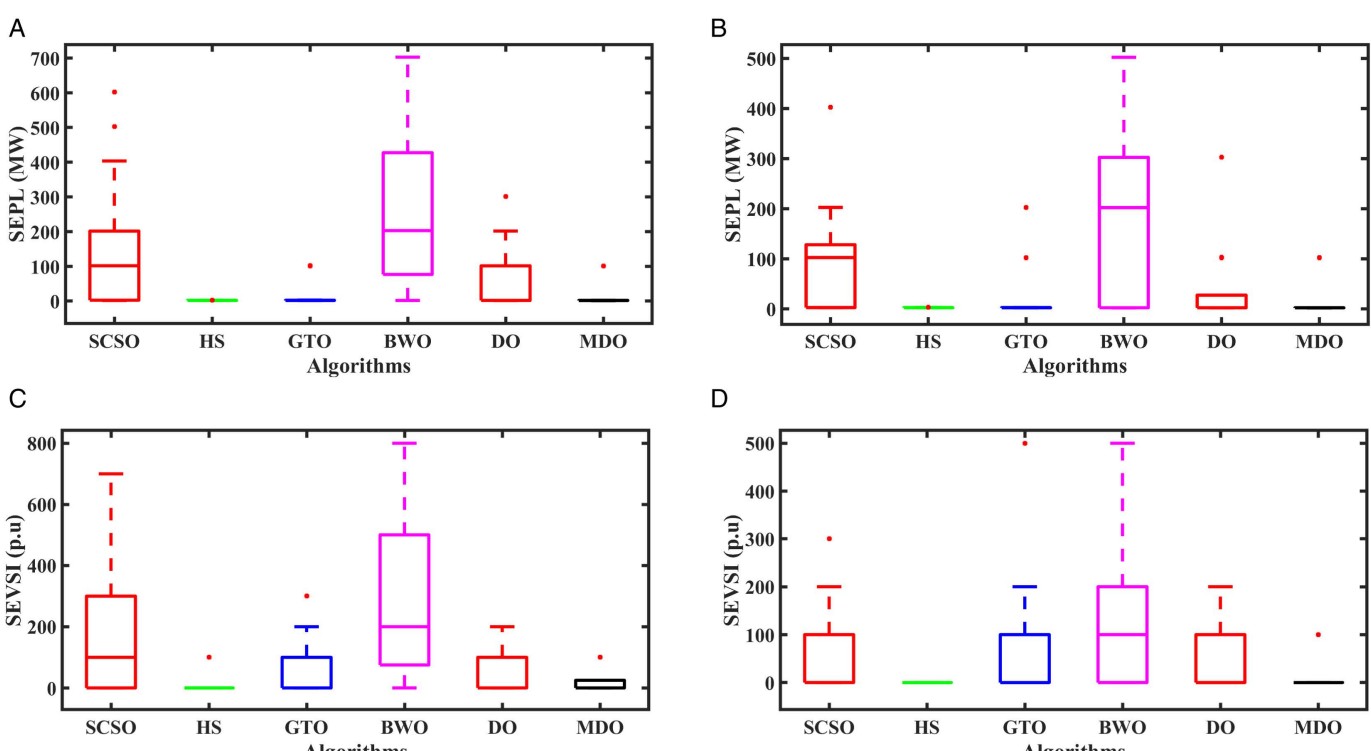

**Fig 12. Classification of objective functions using box plots: (a) SEPL with considering RERs, (b) SEPL without considering RERs, (c) SEVS with considering RERs, and (d) SEVS without considering RERs.**

**Table 6. Sensitivity analysis of SEPL under different number of scenarios.**

| Nombre de scenarios | SEPL | | | | | |
|---|---|---|---|---|---|---|
| | SCSO | HS | GTO | BWO | DO | MDO |
| 5 | 1.47043 | 1.44122 | 1.40720 | 1.51553 | 1.42929 | 1.37529 |
| 10 | 1.54823 | 1.47023 | 1.36967 | 1.58519 | 1.40488 | 1.29221 |
| 15 | 1.84510 | 1.80766 | 1.60410 | 1.98856 | 1.70479 | 1.26277 |

**Table 7. Sensitivity analysis of SEVS under different number of scenarios.**

| Nombre de scenarios | SEVS | | | | | |
|---|---|---|---|---|---|---|
| | SCSO | HS | GTO | BWO | DO | MDO |
| 5 | 0.08770 | 0.08683 | 0.08712 | 0.08716 | 0.08756 | 0.08592 |
| 10 | 0.07247 | 0.06866 | 0.06684 | 0.09085 | 0.07006 | 0.06440 |
| 15 | 0.06816 | 0.06983 | 0.06811 | 0.09958 | 0.06388 | 0.06167 |

**Table 8. Statistical comparison for SEPL, and SEVS for different algorithms in the two case studies for 25 runs.**

| Item | | | mean | Best | Worst | SD | Variance | 95% confidence intervals | Time(s) | PV value |
|---|---|---|---|---|---|---|---|---|---|---|
| SEPL | Without RERs | SCSO | 98.64 | 2.30 | 402.6 | 97.8 | 9564.0 | [58.25, 139.02] | 7930 | **2.45E-07** |
| | | HS | 2.51 | 2.22 | 3.2 | 0.2 | 0.0 | [2.416, 2.597] | 9477 | 7.35E-04 |
| | | GTO | 22.35 | 2.07 | 202.5 | 50.0 | 2502.0 | [1.704, 43.00] | 4472 | 0.01165 |
| | | BWO | 1.90.41 | 2.02 | 502.3 | 176.5 | 31160.0 | [117.52, 263.31] | 4179 | 8.88E-05 |
| | | DO | 34.40 | 2.03 | 302.9 | 69.2 | 4792.0 | [5.826, 62.97] | 3759 | 0.01613 |
| | | MDO | 18.17 | 1.88 | 102.6 | 37.5 | 1404.0 | [2.703, 33.64] | 15764 | – |
| | With RERs | SCSO | 150.31 | 1.85 | 602.5 | 166.2 | 27640.0 | [81.69, 218.93] | 5273 | 3.02E-07 |
| | | HS | 2.15 | 1.81 | 3.2 | 0.3 | 0.1 | [2.032, 2.274] | 5789 | 0.02834 |
| | | GTO | 22.02 | 1.60 | 102.3 | 40.9 | 1671.0 | [5.13, 38.91] | 3692 | 0.10313 |
| | | BWO | 266.40 | 1.99 | 702.7 | 234.3 | 54910.0 | [169.72, 363.09] | 4047 | 6.89E-08 |
| | | DO | 50.03 | 1.70 | 301.9 | 77.1 | 5937.0 | [18.23, 81.83] | 3675 | 0.00434 |
| | | MDO | 9.88 | 1.26 | 102.1 | 27.7 | 768.1 | [0, 21.32] | 14188 | – |
| SEVS | Without RERs | SCSO | 72.10 | 0.10 | 300.1 | 93.6 | 8768.0 | [33.50, 110.71] | 8578 | 1.06E-05 |
| | | HS | 0.10 | 0.10 | 0.1 | 0.0 | 0.0 | [0.1021, 0.1034] | 59690 | 0.02319 |
| | | GTO | 88.10 | 0.10 | 500.1 | 113.0 | 12760.0 | [41.47, 134.73] | 4045 | 5.53E-04 |
| | | BWO | 152.10 | 0.10 | 500.1 | 150.3 | 22600.0 | [90.10, 213.22] | 4527 | 4.10E-07 |
| | | DO | 52.10 | 0.10 | 200.1 | 65.3 | 4267.0 | [25.18, 78.02] | 3795 | 0.00552 |
| | | MDO | 8.10 | 0.09 | 100.1 | 27.7 | 766.9 | [0, 19.51] | 14485 | – |
| | Without RERs | SCSO | 176.09 | 0.07 | 700.1 | 180.9 | 32730.0 | [101.37, 250.81] | 7321 | 1.97E-05 |
| | | HS | 12.09 | 0.07 | 100.1 | 33.2 | 1098.0 | [-1.60, 25.77] | 70455 | 0.16832 |
| | | GTO | 64.09 | 0.07 | 300.1 | 95.2 | 9068.0 | [24.79, 103.39] | 5954 | 0.13017 |
| | | BWO | 280.10 | 0.10 | 800.1 | 256.6 | 65860.0 | [174.20, 386.00] | 9155 | 8.55E-08 |
| | | DO | 52.09 | 0.06 | 200.1 | 71.4 | 5104.0 | [22.61, 81.56] | 7856 | 0.42631 |
| | | MDO | 24.09 | 0.06 | 100.1 | 43.6 | 1895.0 | [6.10, 42.08] | 16852 | – |

generated scenarios. The results obtained by MDO have been compared with BWO, HS, SCSO, GTO and the traditional DO in terms of the statistical comparison. Friedman and Wilcoxon tests, the convergence and the boxplot. As per the results obtained, the enhancement of SEPL and SEVS are 32.82% and 34.64% with incorporating the RERs.

## Author contributions

**Conceptualization:** Naima Agouzoul, Mohamed Ebeed.

**Data curation:** Aziz Oukennou, Mohamed Ebeed.

**Formal analysis:** Mokhtar Aly.

**Funding acquisition:** Emad A. Mohamed.

**Investigation:** Naima Agouzoul, Emad A. Mohamed.

**Methodology:** Naima Agouzoul, Jamal Boukherouaa.

**Resources:** Faissal Elmariami, Aziz Oukennou.

**Software:** Aziz Oukennou, Mohamed Ebeed.

**Supervision:** Mohamed Ebeed, Mokhtar Aly.

**Validation:** Mohamed Ebeed, Rabiaa Gadal.

**Writing – original draft:** Faissal Elmariami, Jamal Boukherouaa.

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
