## [Decision Letter · Decision Letter 0]

PONE-D-25-11087Optimization the Stochastic Optimal Reactive Power Dispatch with renewable energy resources using a Modified Dandelion AlgorithmPLOS ONE

Dear Dr. Ebeed,

Thank you for submitting your manuscript to PLOS ONE. After careful consideration, we feel that it has merit but does not fully meet PLOS ONE’s publication criteria as it currently stands. Therefore, we invite you to submit a revised version of the manuscript that addresses the points raised during the review process.

Based on the reviewers' feedback, a **major revision** is required before further consideration. The Key concerns include clarity in technical language, need for more details on Modified Dandelion Optimizer algorithm, and a refined contribution statement. The scenario selection process and sensitivity analysis need elaboration, along with a detailed discussion on computational complexity, convergence behaviour, and statistical analysis. Figures should be improved. Please address these issues and submit a revised manuscript with a detailed response.

We look forward to receiving your revised manuscript.

Kind regards,

Chandan Kumar Shiva

Academic Editor

PLOS ONE

“This study is supported via funding from Prince sattam bin Abdulaziz University project number (PSAU/2025/R/1446).”

Reviewers' comments:

Reviewer's Responses to Questions

**Comments to the Author**

1. Is the manuscript technically sound, and do the data support the conclusions?

Reviewer #1: Yes

Reviewer #2: Yes

2. Has the statistical analysis been performed appropriately and rigorously? 

Reviewer #1: Yes

Reviewer #2: Yes

3. Have the authors made all data underlying the findings in their manuscript fully available?

Reviewer #1: Yes

Reviewer #2: Yes

4. Is the manuscript presented in an intelligible fashion and written in standard English?

Reviewer #1: Yes

Reviewer #2: Yes

5. Review Comments to the Author

Reviewer #1: 1- The paper contains multiple language issues that make the text hard to follow (e.g., “increases the complicities,” “application the proposed algorithm”). Careful proofreading is needed to improve readability and clarity.

2- Some terms and ideas are not clearly defined, like "stochastic fluctuations" and "expected voltage stability." A clearer definition of technical terms would help readers from broader backgrounds.

3- While the enhancements to the Dandelion Optimizer are interesting, the paper would benefit from a more in-depth discussion of why these specific techniques (QOBL, WFM, FDB) were chosen and how they address limitations of the original Dandelion Optimizer.

4- The contribution of the paper compared to other hybrid optimization approaches needs to be more clearly stated. How does this work advance the state of the art in SORPD beyond existing techniques?

5- The paper mentions 15 scenarios for representing load and RER uncertainties but does not clarify how these scenarios were selected or reduced. More insight into the scenario generation and reduction process would add credibility to the stochastic modeling.

6- A sensitivity analysis on the number of scenarios would strengthen the argument for the robustness of the proposed approach.

7- The paper should discuss the computational complexity of the MDO algorithm and how it scales with larger systems.

Convergence behavior and runtime performance compared to the other optimization techniques should be clearly presented, ideally with convergence plots.

8- While the results seem promising, the paper would benefit from a more detailed discussion of why the MDO algorithm outperforms others. What specific characteristics of the enhanced MDO contribute to its superior performance?

9- Statistical analysis of the results (mean, variance, confidence intervals) would provide a more rigorous comparison between optimization techniques.

10- figure 9 is repeated

11- improve figure 1, 9, 10

Reviewer #2: 1- More details are needed about problem formulation.

2- Title of 3.2 “ Modified Dandelion optimizer “

Title of 4.2 “ Modified DANDELION OPTIMIZER “

are same.

3- Numbers of titles and subtitles should be corrected.

4- At figure 12 , give a reason, why BWO algorithm is the highest value of SEPL (MW) than other algorithms.

6. PLOS authors have the option to publish the peer review history of their article (what does this mean? ). If published, this will include your full peer review and any attached files.

**Do you want your identity to be public for this peer review?** For information about this choice, including consent withdrawal, please see our Privacy Policy .

Reviewer #1: **Yes: ** Khairy Sayed

Reviewer #2: **Yes: ** Loai Saad Eldeen NASRAT

---

## [Author Response · Author response to Decision Letter 1]

2 May 2025

The response to reviewer comments has been uploded

---

## [Editor Report · Decision Letter 1]

PONE-D-25-11087R1Optimization the Stochastic Optimal Reactive Power Dispatch with renewable energy resources using a Modified Dandelion AlgorithmPLOS ONE

Dear Dr. Ebeed,

Thank you for submitting your manuscript to PLOS ONE. After careful consideration, we feel that it has merit but does not fully meet PLOS ONE’s publication criteria as it currently stands. Therefore, we invite you to submit a revised version of the manuscript that addresses the points raised during the review process.

<h1>Editor's Decision: Minor Revision</h1>

Based on the reviewers' comments, this manuscript requires minor revisions before publication. Both reviewers identified issues with formatting, language clarity, and technical content. Key improvements needed include proofreading for language errors, clearer definitions of technical terms, justification for the proposed algorithm modifications, more rigorous comparison with existing techniques, detailed analysis of computational complexity, and correction of formatting inconsistencies and figure issues. Please address the duplicate Figure 9, improve Figures 1, 9, and 10, and explain the BWO algorithm's higher SEPL value.

We look forward to receiving your revised manuscript.

Kind regards,

Chandan Kumar Shiva

Academic Editor

PLOS ONE
---

## [Author Response · Author response to Decision Letter 2]

19 Jun 2025

Comment 1: “Both reviewers identified issues with formatting, language clarity, and technical content.”

Authors’ Response: The authors appreciate the editor and reviewers’ comments. A thorough proofreading was carried out. Linguistic and grammatical errors were corrected. Additional explanations have been added to clarify ambiguous terms. The formatting of the manuscript has been revised.

Comment 2: “justification for the proposed algorithm modifications “

Authors’ Response: The authors appreciate the editor and reviewers’ comments. The limitations of the original Dandelion Optimizer (DO) are that it prone to local optima and suffers from stagnation in case of solving the nonlinear objective functions like ORPD. In this regard, three modifications have been integrated for improving the searching ability of the DO using FDB, QOBL, and WFM.

Comment 3: “More rigorous comparison with existing techniques “

Authors’ Response: The authors appreciate the editor and reviewers’ comments. A comparsion between the proposed algorithm and the other techniques has included in Table 8 in the revised paper.

Comment 4: “detailed analysis of computational complexity “

Authors’ Response: The authors appreciate the editor and reviewers’ comments.” It should be mentioned that the proposed MDO has a high level of computational complexity, because the DO is based on three stages, including rising, landing, and descending. Hence, the computational complexity of the DO is O(M×T×pop×Dim×fit), where M, T, pop, Dim, and fit are the current optimal solution, the maximum iteration number, the number of populations, the problem dimension, and the objective function [87]. The MDO is based on three modifications, including the WFM, QOBL, and FDB, Consequently, the computational complexity of the MDO is O(M×T×pop×Dim×fit +3(M×T×pop×Dim×fit)). Hence, the computational complexity of the MDO is O(4(M×T×pop×Dim×fit)).” This explanation is added on the manuscript between lines 508 and 515.

Comment 5: “Figure 9 is repeated.”

Authors’ Response: The authors appreciate the editor and reviewers’ comments. Figure 9 has been updated to avoid repetition.

Comment 6: “improve figure 1, 9, 10.”

Authors’ Response: The authors appreciate the editor and reviewers’ comments. The quality of the mentioned figures has been improved on the manuscript.

Comment 7: “Please explain the BWO algorithm's higher SEPL value. “

Authors’ Response: The authors sincerely appreciate the editor's and reviewers’ insightful comments.

“The BWO algorithm distributes its candidate solutions widely in the search space to avoid premature convergence and accurately explore potential optimal solutions. While this intensive exploration allows local minima to be avoided, it can also lead to less consistent or higher values of the objective function in some runs. It is important to note that Figure 12 shows the distribution of the objective function, which includes both the objective function and the system constraints.

---

## [Editor Report · Decision Letter 2]

Optimization the stochastic optimal reactive power dispatch with renewable energy resources using a modified dandelion algorithm

PONE-D-25-11087R2

Dear Dr. Ebeed,

We’re pleased to inform you that your manuscript has been judged scientifically suitable for publication and will be formally accepted for publication once it meets all outstanding technical requirements.

Kind regards,

Chandan Kumar Shiva

Academic Editor

PLOS ONE
---

## [Editor Report · Acceptance letter]

PONE-D-25-11087R2

PLOS ONE

Dear Dr. Ebeed,

I'm pleased to inform you that your manuscript has been deemed suitable for publication in PLOS ONE. Congratulations! Your manuscript is now being handed over to our production team.

Kind regards,

on behalf of

Dr. Chandan Kumar Shiva

Academic Editor

PLOS ONE